# Scaling Embedding Layers in Language Models

**Da Yu**[†]    **Edith Cohen**[†]    **Badih Ghazi**[†]    **Yangsibo Huang**[†]    **Pritish Kamath**[†]

**Ravi Kumar**[†]       **Daogao Liu**[†]       **Chiyuan Zhang**[†]

## Abstract

We propose SCONE (**S**calable, **C**ontextualized, **O**ffloaded, **N**-gram **E**mbedding), a new method for extending input embedding layers to enhance language model performance. To avoid increased decoding costs, SCONE retains the original vocabulary while introducing embeddings for a set of frequent $n$-grams. These embeddings provide contextualized representation for each input token and are learned with a separate model during training. After training, embeddings are precomputed and stored in off-accelerator memory; during inference, querying them has minimal impact on latency due to the low complexity of embedding lookups. SCONE enables two new scaling strategies: increasing the number of $n$-gram embeddings and scaling the model used to learn them, both while maintaining fixed accelerator usage during inference (in terms of FLOPS and memory). We show that scaling both aspects enables a model with 1B accelerator-resident parameters to outperform a 1.9B-parameter baseline across diverse corpora, while using only about half the FLOPS and accelerator memory during inference.

## 1 Introduction

Input embedding layers in language models map discrete tokens to continuous vector representations [Mikolov et al., 2013, Sennrich et al., 2016] before passing them to the subsequent layers. Since tokens are typically simple integer values, the mapping can be implemented purely as memory fetch operations with no additional computation needed. This allows embedding layers to be offloaded from the limited accelerator memory to main memory or even to secondary storage, such as solid-state drives, with minimal impact on inference speed. This is highly desirable, as main memory and secondary storage are significantly more cost-effective than accelerator memory (e.g., GPUs and TPUs) [McCallum, 2024]. These advantages motivate us to explore methods for scaling up embedding layers.

However, scaling the embedding layer by simply increasing the token vocabulary size has limited benefits. A larger vocabulary also enlarges the output (logits) layer, whose weight matrix is tied to the vocabulary size and embedding dimension. Typically, predicting the next token requires computing logits over all tokens in the vocabulary, and prior work shows that the decoding cost becomes impractical once the vocabulary exceeds a few hundred thousand tokens [Wang et al., 2019, Zheng et al., 2021, Liang et al., 2023, Tao et al., 2024, Dagan et al., 2024]. Even if faster hardware or more efficient algorithms might partially offset this cost [Joulin et al., 2017, Shim et al., 2017], a second problem remains: scaling the token vocabulary leads to a large number of *tail tokens*, which occur infrequently in the training corpus. The embeddings of these tokens (both input and output) receive very few updates, resulting in lower-quality representations [Liao et al., 2021, Dou et al., 2024]. In Appendix D, we train GPT-2 models with vocabulary sizes ranging from 32K to 2M and observe performance degradation as the vocabulary size exceeds 512K. We attribute this degradation

---

[†]Google. `Correspondence to:` {`dayuwork,edco,pritishk,chiyuan`}`@google.com`.

39th Conference on Neural Information Processing Systems (NeurIPS 2025).

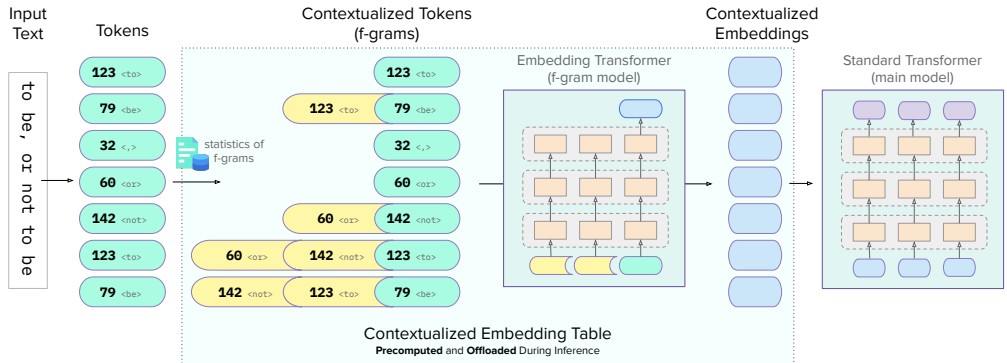

Figure 1: Illustration of SCONE with a maximum $n$-gram length of 3. The *f-grams* are a set of frequent $n$-grams selected using the method described in Section 3.1.

to the increasing sparsity of updates per token as the vocabulary grows. Additionally, we observe a linear increase in accelerator memory usage once the vocabulary size exceeds 1M.

**Our contributions.** In this paper, we propose SCONE, a novel approach for scaling input embedding layers by learning them through a separate transformer model, referred to as the *f-gram model*. This model takes as input a set of frequently occurring $n$-grams (called *f-grams*), which we select using an approach inspired by Byte-Pair Encoding-based tokenization (Section 3.1). Crucially, the number of f-grams is decoupled from the token vocabulary size, allowing us to build a separate f-gram input embedding table with up to billions of entries, without blowing up the vocabulary size. An overview of the proposed method is illustrated in Figure 1.

During both training and inference, SCONE leverages the f-gram model to efficiently handle large embedding spaces without overwhelming accelerator resources. During training, the f-gram model learns to generate contextualized embeddings for each f-gram without requiring the instantiation of a massive embedding table. During inference, the output of the f-gram model can be precomputed to form the *f-gram embedding layer*. This embedding layer can then be offloaded from the accelerator, thereby eliminating the need for accelerator resources during inference.

SCONE introduces two novel scaling approaches for improving model performance: (i) increasing the number of cached f-gram embeddings and (ii) scaling up the f-gram model used to learn these embeddings. The first approach requires additional off-accelerator memory during inference, while the second demands greater accelerator resources during training. Importantly, both approaches preserve a fixed inference-time accelerator resource footprint, a property *not* supported by traditional scaling methods. Indeed, prior works [Jones, 2021, Hoffmann et al., 2022] have shown that scaling the training compute for a fixed model size beyond some compute-optimal threshold leads to diminishing returns. Therefore, the typical method for utilizing additional training compute is to increase model size. However, directly scaling model size also increases FLOPS and accelerator memory requirements during inference. In contrast, SCONE leverages larger f-gram models to effectively consume more accelerator resources during training but without increasing inference-time accelerator demands, offering a novel scaling paradigm that previous studies have not explored.

There are important scenarios where maintaining a fixed inference-time accelerator footprint is especially valuable. In many deployments, where a model is queried billions of times per day, inference costs during deployment can far exceed training costs. In such cases, even small increases in inference-time computation can lead to substantial operational expenses. The recent emergence of test-time scaling techniques further highlights this trend [Jones, 2021, Snell et al., 2025], emphasizing situations where inference costs dominate the overall expenses of LLM deployment. Furthermore, many latency-sensitive applications impose strict limits on inference-time computation, leaving no margin for increased computational demands during deployment.

Our contributions can be summarized as follows:

- A new scalable method, SCONE, to improve language models by expanding the input embeddings (Section 3) but requiring no additional accelerator resources at inference time.
- Extensive experiments to validate SCONE, analyzing key design choices and their impact on evaluation perplexity and accuracy on downstream tasks (Section 4).

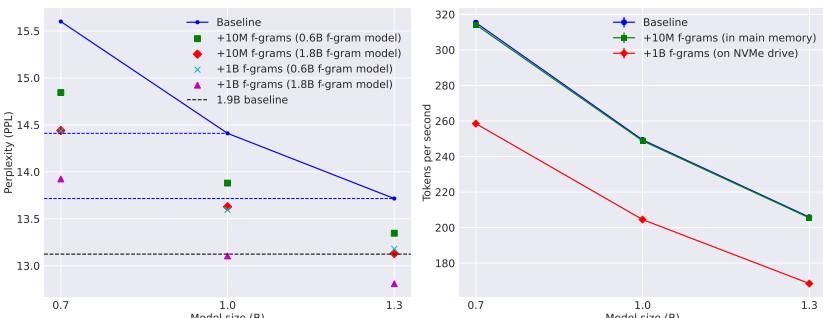

Figure 2: **(Left)** Perplexity on the OLMo [Groeneveld et al., 2024] evaluation set. Model sizes along the $x$-axis indicate the number of parameters residing on the accelerator during inference. With 10M f-grams, the 1.3B model matches the performance of the 1.9B baseline; with 1B f-grams, the 1B model surpasses it. **(Right)** End-to-end token generation speed on a single A100 GPU. Storing f-gram embeddings in main memory adds negligible latency, while using NVMe storage introduces a minor slowdown without causing a bottleneck.

Our results show that SCONE significantly boosts model performance without introducing inference latency bottlenecks; Figure 2 highlights representative findings. Notably, a SCONE model with 1B accelerator-resident parameters outperforms a 1.9B baseline that requires approximately $2\times$ more inference FLOPS and accelerator memory.

## 2 Preliminaries

We focus on pre-training decoder-only language models with causal language modeling [Radford et al., 2019]. We now introduce notations that will later help us formally describe the proposed SCONE method. For clarity, we omit details that are not essential to describing our method.

**Decoder Transformer Model.** Let $V_{\text{token}}$ denote the token vocabulary. The *token embedding layer* is parameterized by a function $\mathcal{T} : V_{\text{token}} \to \mathbb{R}^d$, mapping each token to a $d$-dimensional embedding vector. We abstract the transformer itself as a function $\mathcal{A} : (\mathbb{R}^d)^{\leq S} \to \mathbb{R}^d$, which takes as input a sequence of up to $S$ token embeddings and outputs a single embedding vector[2]. For completeness, we present the pseudocode of a basic next-token prediction model in Algorithm 2 (Appendix C).

**Efficient Indexing for Large Embedding Layers.** Mappings from tokens or f-grams to embedding vectors can be implemented as key–value stores, enabling highly efficient lookup. Hash-based and tree-based data structures support lookup times that are constant or logarithmic in the number of entries, respectively. These data structures make it feasible, in principle, to offload large embedding layers from accelerators with minimal impact on latency. In practice, however, the traditional token embedding layer $\mathcal{T}$ is kept on accelerator memory, as it is typically shared with the output (logits) layer, which requires fast access for matrix multiplications during next-word prediction. In contrast, the f-gram embedding layer introduced by SCONE is decoupled from the output layer and can therefore be offloaded, which is critical for maintaining a fixed accelerator memory footprint as the f-gram layer scales. In Section 4.3, we study two strategies for storing the f-gram embedding table. When stored in main system memory, the f-gram embedding table consists of a dense embedding matrix paired with a hash dictionary that maps f-grams to matrix indices. When stored on NVMe solid-state drives, we use the Lightning Memory-Mapped Database (LMDB) [Chu, 2011] to directly map f-grams to their embeddings using a B+ tree data structure.

## 3 SCONE Architecture

We propose to augment a standard transformer model with an additional f-gram embedding layer. Figure 1 provides a high-level overview of the approach. We first construct a set $V_{\text{f-gram}} \subseteq V_{\text{token}}^{[2,K]} := \bigcup_{n=2}^{K} V_{\text{token}}^n$ consisting of frequently occurring $n$-grams of length up to $K$, which we refer to as *f-grams*. Throughout this paper, $K$ denotes the maximum length of f-grams considered. To construct

---

[2]A decoder transformer typically produces a sequence of embeddings for an input sequence. We define the output as the final token embedding, used either for next-word prediction or as the embedding for a f-gram.

**Algorithm 1** SCONE method $F_{\mathcal{T}, V_{\text{f-gram}}, \mathcal{A}_{\text{f-gram}} | \mathcal{F}}$.

- $\mathcal{T} : V_{\text{token}} \to \mathbb{R}^d$: token embedding layer.
- $V_{\text{f-gram}} \subseteq V_{\text{token}}^{\leq K}$: set of f-grams.
- **TRAINING**: f-gram transformer model
  - $\triangleright\ \mathcal{A}_{\text{f-gram}} : (\mathbb{R}^d)^{\leq K} \to \mathbb{R}^d$.
- **INFERENCE**: f-gram embedding layer
  - $\triangleright\ \mathcal{F} : V_{\text{f-gram}} \to \mathbb{R}^d$.

**Input:** A sequence $(\sigma_1, \ldots, \sigma_m) \in V_{\text{token}}^m$ of tokens.
**Output:** Embeddings $(\boldsymbol{e}_1, \ldots, \boldsymbol{e}_m) \in (\mathbb{R}^d)^m$.
**for** $i = 1, \ldots, m$ **do**
    $j \leftarrow$ smallest $j' < i$ such that $(\sigma_{j'}, \ldots, \sigma_i) \in V_{\text{f-gram}}$ if such a $j'$ exists, otherwise $i$.
    **if** $j = i$ **then**
        $\boldsymbol{e}_i \leftarrow \mathcal{T}(\sigma_i)$
    **else**
$$\boldsymbol{e}_i \leftarrow \begin{cases} \mathcal{A}_{\text{f-gram}}(\mathcal{T}(\sigma_j), \ldots, \mathcal{T}(\sigma_i)) & \text{at training,} \\ \mathcal{F}(\sigma_j, \ldots, \sigma_i) & \text{at inference.} \end{cases}$$
**return** $(\boldsymbol{e}_1, \ldots, \boldsymbol{e}_m)$

$V_{\text{f-gram}}$, we use an efficient implementation that requires only $K - 1$ linear scans over the training corpus, as detailed in Section 3.1; the formal construction is described in Algorithm 3 (Appendix C).

Next, we define the SCONE method, which maps a given sequence of tokens to a sequence of embeddings. SCONE behaves differently during training and inference, as described in Algorithm 1. During training, it is parameterized by an f-gram transformer model $\mathcal{A}_{\text{f-gram}} : (\mathbb{R}^d)^{\leq K} \to \mathbb{R}^d$, which takes an f-gram as input and uses the output embedding of the final token as the embedding for that f-gram. During inference, in contrast, it is parameterized by an f-gram embedding layer $\mathcal{F} : V_{\text{f-gram}} \to \mathbb{R}^d$, a key–value store that directly maps each f-gram to its precomputed embedding. This embedding layer is implemented by caching the outputs of $\mathcal{A}_{\text{f-gram}}$ for all f-grams in $V_{\text{f-gram}}$ and storing them in off-accelerator memory.

The embeddings produced by SCONE are passed to a standard transformer model $\mathcal{A}_{\text{main}}$, referred to as the *main model*. This is followed by a prediction head $\mathcal{D} : \mathbb{R}^d \to \Delta_{V_{\text{token}}}$. Together, these components form the end-to-end process for next-word prediction with SCONE. We present the full description in Algorithm 4 (Appendix C).

In the rest of this section, we will discuss the motivation behind the design chocies of SCONE and provide further implementation details.

### 3.1 BPE-Style Discovery of f-grams

The construction of $V_{\text{f-gram}}$, outlined in Algorithm 3 (Appendix C), can be implemented efficiently with $K - 1$ linear scans over the training corpus. We perform one scan for each $n \in [2, K]$, starting with 2-grams. In subsequent scans, we impose a minimum frequency threshold of 5 to reduce memory usage. At the $(n + 1)$th scan, the set of $n$-grams from the previous scan allows us to skip any $(n + 1)$-gram candidates that cannot meet the minimum threshold. Specifically, if an $(n + 1)$-gram surpasses the threshold, its $n$-suffix or prefix must appear at least as many times. Figure 3 shows how the number of unique $n$-grams (up to 6-grams) grows as the training corpus scales from a few billion to one trillion tokens. Finally, all found $n$-grams (for $n \in [2, K]$) are ranked by frequency, and the top ones are selected to comprise $V_{\text{f-gram}}$.

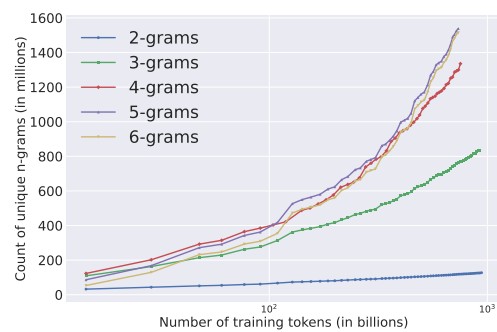

Figure 3: Number of unique 2- to 6-grams appearing at least five times. We uniformly sample tokenized sequences from Dolma [Soldaini et al., 2024] to vary the corpus size.

Our procedure for counting and ranking $n$-grams is analogous to continuing the training of a BPE tokenizer on an existing vocabulary. In each BPE iteration [Gage, 1994, Sennrich et al., 2016], the frequencies of all token pairs (2-grams) are counted, and the most frequent pair is merged to form a new token, expanding the vocabulary by one. However, merging and recounting pairs repeatedly to obtain a large number of f-grams is prohibitively expensive for large training corpora. Instead, we simply collect and sort all $n$-grams up to a small constant $K$.

## 3.2 Learning f-gram Embeddings with $\mathcal{A}_{\text{f-gram}}$

We motivate the use of $\mathcal{A}_{\text{f-gram}}$ by contrasting it with the alternative of directly backpropagating gradients to a large embedding table. The direct approach fails to exploit dependencies between $n$-grams, leading to fewer updates per embedding. We observed this by pretraining GPT-2 models with vocabulary sizes ranging from 32K to 2M. As vocabulary size increases, embedding updates become sparser, eventually degrading performance. For example, when training on 100M tokens, 97.6% of tokens in a 32K vocabulary receive more than 100 updates, compared to only 7.3% in a 2M vocabulary (Appendix D). This sparsity makes it difficult to train a large embedding table through direct gradient updates. SCONE addresses this by parameterizing embeddings with an f-gram transformer $\mathcal{A}_{\text{f-gram}}$, avoiding the sparse update problem.

Additionally, $\mathcal{A}_{\text{f-gram}}$ removes the need to instantiate a full embedding table during training, a requirement that would otherwise strain accelerator memory. This is because, unlike inference, where next-word prediction is largely sequential, training parallelizes computation across the sequence dimension, demanding much higher token throughput. Moreover, during training, embeddings must be updated frequently, whereas inference only requires read access. Together, these factors make it difficult to offload the embedding layer from accelerators during training. As a result, avoiding full-table instantiation is crucial for scaling the embedding layer to extremely large sizes.

SCONE jointly trains the $\mathcal{A}_{\text{f-gram}}$ model with the main model $\mathcal{A}_{\text{main}}$ and the token embedding layer $\mathcal{T}$. This overcomes the sparse updates issue but also introduces additional compute costs. For each $\omega \in V_{\text{f-gram}}$, the computation is the same as that of processing a short sequence of length $|\omega|$ through a standard transformer. Since $|\omega|$ is a small constant, the primary overhead comes from the feed-forward layer. In our experiments (Section 4), we account for this overhead in one of two ways: (1) by training baseline models long enough to reach near-convergence at a fixed model size, ensuring that additional training compute would yield minimal gains; or (2) by reducing the number of training tokens for models using SCONE so that their total training FLOPS match those of the baselines.

During inference, the f-gram embedding layer $\mathcal{F}$ can be precomputed and stored in a lookup table, offloaded to system main memory or secondary storage while still permitting efficient retrieval. Meanwhile, the token embedding layer $\mathcal{T}$ remains on the accelerator for decoding. In Section 4.3, we evaluate the query latency and space usage of the f-gram embedding layer under various configurations. We show that the latency is not a bottleneck for language model inference and the space costs are low due to the use of relatively inexpensive system memory and solid-state drives.

## 4 Experimental Evaluation

In this section, we evaluate SCONE in pre-training settings. In Section 4.1, we assess SCONE on GPT-2–sized models to study various design choices, and in Section 4.2, we extend the evaluation to large-scale pre-training scenarios involving trillions of tokens. Finally, in Section 4.3, we analyze the inference and storage costs during deployment.

For completeness, we also evaluate SCONE in post-training settings by applying it during the SFT stage of recent Qwen3 models [Yang et al., 2025]; these results, presented in Appendix E.3, show that SCONE remains effective in post-training as well.

### 4.1 Experiments with GPT-2

We analyze three key hyperparameters: (i) the maximum f-gram length $K$ in $V_{\text{f-gram}}$, (ii) the number of f-grams used, $|V_{\text{f-gram}}|$, and (iii) the $\mathcal{A}_{\text{f-gram}}$ model size. We use the released GPT-2 tokenizer, which has $|V_{\text{token}}| = 50{,}257$, and train on the WebText dataset [Peterson et al., 2019]. The tokenized corpus contains 9B training tokens, from which we extract f-grams using the method in Section 3.1.

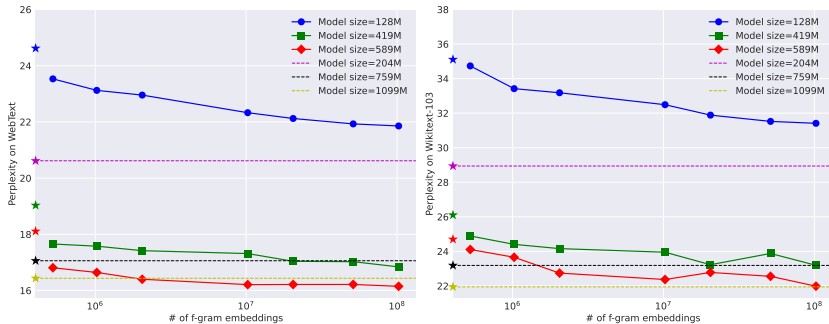

Figure 4: Evaluation perplexity on WebText (left) and WikiText-103 (right) as a function of $|V_{\text{f-gram}}|$. Model sizes in the legend are number of parameters residing on the accelerator during inference. Dashed lines and leftmost stars show baseline performance.

We consider three main model sizes with 76M, 340M, and 510M non-embedding parameters. Including the token embedding layer, the total parameter counts increase to 128M, 419M, and 589M, respectively. These models are either trained using only the token embedding layer as a baseline or with an additional $\mathcal{A}_{\text{f-gram}}$ when SCONE is applied. We train all models for 80B tokens, roughly twice the number of training tokens used in Radford et al. [2018], to ensure that the baseline models approach convergence. For evaluation, we use the validation split of WebText and WikiText-103 [Merity et al., 2017], one of the largest downstream datasets in Radford et al. [2019]. Additional implementation details are provided in Appendix F.1.

### 4.1.1 Varying the Maximum f-gram Length

We study the impact of varying the maximum f-gram length $K$ in $V_{\text{f-gram}}$. We vary $K$ from 2 to 8 while keeping the total number of f-grams fixed at 20M. As $K$ increases, the frequency cutoff, which is the minimum number of times an $n$-gram appears in the training corpus to be included in $V_{\text{f-gram}}$, also increases. The cutoff is 7 when $K = 2$ and 108 when $K = 8$.

For each value of $K$, we evaluate (i) the model's perplexity and (ii) the average length of matched f-grams on WikiText-103. The results are shown in Figure 5. We find that evaluation perplexity rises between $K = 2$ and $K = 4$, after which it plateaus with some fluctuations. A similar trend is observed for the average match length, defined as the average length of the f-grams matched by SCONE for each token in the evaluation set. The average match length also increases between $K = 2$ and $K = 4$ and then stabilizes. This trend is likely because longer f-grams are rarer after frequency-based ranking. As a result, even when $K$ is larger, most selected $n$-grams remain short. Additionally, longer f-grams from the training corpus are less likely to match sequences in downstream data. Experiments on the WebText validation split (Appendix E.1) show a similar trend, although the average match length continues to increase slightly longer, plateauing around $K = 6$.

Considering these findings, for the experiments in the remainder of this paper, we set the maximum f-gram length to $K = 5$ unless stated otherwise.

### 4.1.2 Varying the Number of f-grams

We observe consistent improvements in language modeling performance as we scale up $|V_{\text{f-gram}}|$. To implement $\mathcal{A}_{\text{f-gram}}$, we replicate the baseline model architecture but remove the token embedding layer. This results in the size of $\mathcal{A}_{\text{f-gram}}$ matches the baseline model's non-embedding parameters.

Figure 4 shows the evaluation perplexity as $|V_{\text{f-gram}}|$ increases from 512K to 100M. On the WebText validation split, the perplexity decreases consistently as the number of f-gram embeddings increases. Similarly, on WikiText-103, the perplexity generally decreases with more f-gram embeddings, though minor fluctuations are observed.

In Figure 4, we include three additional baselines where the non-embedding parameters of the three main models are doubled, resulting in models with 204M, 759M, and 1099M parameters for the original 128M, 419M, and 589M models, respectively. This ensures that the total parameter count of each baseline matches the training-time parameter count when SCONE is applied. With 100M f-gram embeddings, the 419M and 589M models trained with SCONE match or surpass the performance

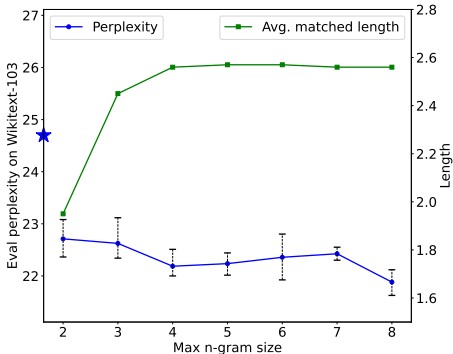

Figure 5: Effect of the maximum f-gram length $n$ on perplexity and matched length. Perplexity decreases as the maximum length increases from 2 to 4, then plateaus with minor fluctuations. Similarly, the average matched length stabilizes after length 4.

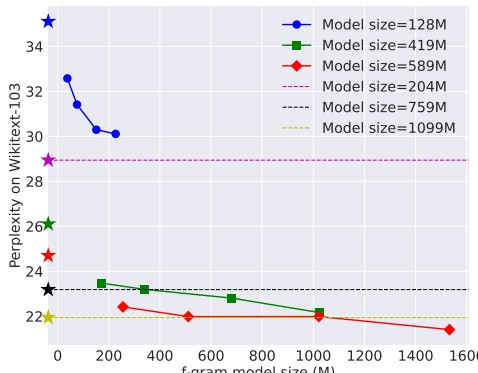

Figure 6: Evaluation perplexity improves as the size of $\mathcal{A}_{\text{f-gram}}$ grows. Model sizes in the legend are main model sizes, including the token embedding layer. Dashed lines and stars on the left represent baselines without an $\mathcal{A}_{\text{f-gram}}$ model.

Table 1: Zero-shot evaluation accuracy on downstream tasks. Models with f-gram embedding layers show clear improvements over the baseline models.

| Model | PIQA | HellaSwag | ARC-E | ARC-C | CSQA | MMLU | Avg. |
|---|---|---|---|---|---|---|---|
| OLMo-1B | 73.6 | 60.9 | 69.5 | 31.8 | 48.7 | 37.6 | 53.7 (+0) |
| OLMo-1.9B | 75.3 | 65.9 | 74.2 | 36.8 | 49.7 | 38.6 | 56.8 (+3.1) |
| OLMo-1B + OE-12.8M [Huang et al., 2025] | 73.7 | 62.7 | 70.3 | 32.1 | 49.9 | 37.8 | 54.4 (+0.7) |
| OLMo-1B + 10M f-grams | 74.0 | 63.6 | 70.4 | 32.1 | 49.9 | 39.3 | 54.9 (+1.2) |
| OLMo-1.3B + 10M f-grams | 75.0 | 65.5 | 75.3 | 36.4 | 49.9 | 38.5 | 56.8 (+3.1) |
| OLMo-1B + 1B f-grams | 75.3 | 67.1 | 72.5 | 36.4 | 50.8 | 39.9 | 57.0 (+3.3) |

of the 759M and 1099M baselines, respectively, despite using only half as many non-embedding parameters during inference.

### 4.1.3   Varying the Size of the $\mathcal{A}_{\text{f-gram}}$ Model

We observe that, for a fixed $|V_{\text{f-gram}}|$, scaling up the $\mathcal{A}_{\text{f-gram}}$ model size provides a new way to improve language modeling performance. We vary the model size by changing the number of layers in the main model architecture. For each $\mathcal{A}_{\text{main}}$ model size, we evaluate four $\mathcal{A}_{\text{f-gram}}$ model sizes: 0.5x, 1x, 2x, and 3x the non-embedding parameters of the main model. We set $|V_{\text{f-gram}}|$ to be 100M. Figure 6 presents the evaluation perplexity on Wikitext-103. The observations on WebText validation split are similar, and we present the results in Appendix E.1.

The results in Figure 6 show that the perplexity generally decreases as the $\mathcal{A}_{\text{f-gram}}$ model size increases, although the improvements become smaller as the model size grows larger. For instance, with the 419M main model, a 170M $\mathcal{A}_{\text{f-gram}}$ model improves the perplexity from 26.1 to 23.4, outperforming the 589M baseline (24.7) by a clear margin. Further scaling of the $\mathcal{A}_{\text{f-gram}}$ model to 1020M (resulting in 1439M total parameters during training) lowers the perplexity to 22.1, which is slightly higher than the 1099M baseline (21.9). This suggests that scaling up the $\mathcal{A}_{\text{f-gram}}$ model initially yields a better scaling curve, but beyond a certain size, it becomes less optimal compared to directly scaling up $\mathcal{A}_{\text{main}}$. However, scaling $\mathcal{A}_{\text{main}}$ also increases accelerator usage during inference, whereas scaling $\mathcal{A}_{\text{f-gram}}$ does not, since it is replaced with an off-accelerator lookup table at inference time. This highlights our method as a novel way to leverage additional training compute while maintaining fixed accelerator usage during inference.

### 4.2   Scaling Up the Training Corpus

After exploring several design choices for SCONE, we now evaluate its performance in large-scale pretraining. Our implementation builds on the open-source OLMo codebase [Groeneveld et al., 2024], licensed under Apache 2.0. Additional implementation details are provided in Appendix F.2.

**Downstream Tasks.** We report zero-shot accuracy on six standard downstream benchmarks: MMLU-var, Hellaswag, ARC-Challenge, ARC-Easy, CommonsenseQA (CSQA), and PIQA. In the main text, we focus on presenting downstream accuracy results under our primary training setting. Additional results, including perplexity evaluations, training curves, and further SCONE configurations under alternative settings, are provided in Appendix E.2. Notably, perplexity trends closely correlate with downstream performance.

**Baselines.** We compare SCONE against three baselines: OLMo-1B, OLMo-1.9B, and a concurrent method, the over-tokenized transformer [Huang et al., 2025]. Model configurations for OLMo-1B and OLMo-1.9B are detailed in Appendix F.2. For the over-tokenized transformer, we adopt the best-performing OE-12.8M variant, which introduces an additional input embedding layer comprising 12.8M embedding vectors, applied on top of the OLMo-1B model. Further discussion of this method can be found in Section 5. All baseline models are trained for 1T tokens.

**SCONE Configuration.** We apply SCONE with two f-gram embedding layer sizes: 10M and 1B f-grams, respectively. The cutoff frequencies are 21,956 for 10M f-grams and 70 for 1B f-grams. SCONE is integrated into OLMo-1B and OLMo-1.3B models, with an $\mathcal{A}_{\text{f-gram}}$ model size of 1.8B parameters (matching the architecture of OLMo-1.9B but excluding the token embedding layer). Models equipped with SCONE are trained for 500B tokens, half the number of tokens used by the baselines, to account for the additional training cost of the $\mathcal{A}_{\text{f-gram}}$ model and ensure comparable total training FLOPS. In Appendix E.2, we also explore a smaller $\mathcal{A}_{\text{f-gram}}$ model of 0.6B parameters, where we observe consistent improvements.

**Results.** Table 1 presents the downstream accuracy results. Models with SCONE demonstrate clear improvements over the baselines. Specifically, with 10M f-grams, OLMo-1.3B achieves parity with the OLMo-1.9B baseline while using approximately 32% less inference FLOPS and accelerator memory. With 1B f-grams, OLMo-1B slightly outperforms the OLMo-1.9B baseline while requiring approximately 48% less inference FLOPS and accelerator memory. Compared to the over-tokenized transformer, OLMo-1B + 10M f-grams surpasses OLMo-1B + OE-12.8M, which we attribute to the additional capacity introduced by the $\mathcal{A}_{\text{f-gram}}$ model during training.

## 4.3 Space Usage and Query Latency

We show that the latency of the f-gram embedding layer does not constitute a bottleneck during inference, and that system memory and solid-state storage are relatively inexpensive. In Appendix E.4, we also provide a unified table summarizing the key metrics: (1) GPU memory, (2) CPU memory, (3) disk usage, and (4) FLOPS/latency for training and inference.

Table 2: Space usage of the f-gram embedding layer $\mathcal{F}$, along with unit cost for memory and NVMe solid-state drives [McCallum, 2024].

| # of $n$-grams | System memory (GB) | Solid-state drive (GB) |
|---|---|---|
| $10^7$ | 41.4 | 77.3 |
| $10^8$ | 413.6 | 766.8 |
| $10^9$ | (does not fit) | 7665.4 |
| Price (per GB) | $\sim 2$ USD | $\sim 0.1$ USD |

We experiment with $|V_{\text{f-gram}}|$ being 10M, 100M, and 1B with embedding dimension of $d = 2048$ and 16-bit precision per floating point value. Experiments were conducted on a workstation with 64 Intel Xeon CPU cores and 512 GB of memory. Space and latency were measured for both in-memory and on-disk storage. In memory, embeddings are stored as a single matrix with a hash dictionary mapping f-grams to indices, while on-disk storage uses the Lightning Memory-Mapped Database [Chu, 2011] to directly store f-gram and embedding pairs on NVMe solid-state drives.

Table 2 summarizes the space usage for both storage methods. In both cases, the space required increases linearly with the number of embedding vectors. The 10M and 100M f-gram embedding layers are able to fit within main memory, with the 10M layer requiring 41.4 GB. For on-disk storage, there is additional overhead as the same 10M layer occupies 77.3 GB storage.

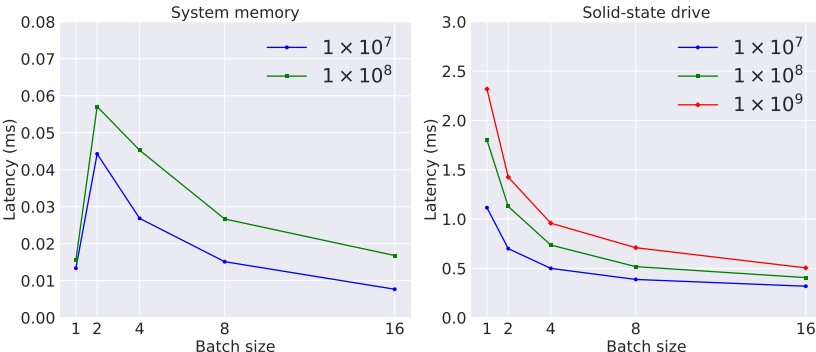

Figure 7: Amortized per-token query latency (ms), averaged over 100,000 batches. The latency spike from batch size 1 to 2 when reading from system memory is due to batch operator overhead, which is less pronounced for solid-state drives.

Figure 7 shows the latency of retrieving embeddings at different batch sizes. Latency is measured as the end-to-end time from loading a batch of tokens to having the f-gram embeddings ready on the GPU. Before each test, the CPU cache is cleared. Up to four queries per token are performed to find the longest matching $n$-gram (for a maximum $n$-gram length of $K = 5$). For in-memory storage, sequential queries are sufficient since they are not the bottleneck, whereas on-disk storage uses parallel queries to the database. At a batch size of 1, retrieval from a 10M f-gram embedding layer on NVMe takes 1.1 ms, increasing to 2.3 ms for a 1B-layer: both well below the latency threshold for LLM inference (typical commercial APIs generate at ~100 tokens/sec, or ~10 ms/token [ArtificialAnlys, 2025]). Larger batch sizes further improve efficiency: at batch size 16, amortized per-token latency drops to 0.5 ms. In-memory access is much faster: for a 100M f-gram layer at batch size 16, per-token latency is only 0.017 ms. We also report end-to-end generation speed in Figure 2, which aligns with these latency trends: when embeddings are stored in main memory, there is negligible impact on throughput; when stored on NVMe drives, throughput slightly decreases but no major bottleneck arises.

## 5  Related Work

**Scaling of Embedding Layers.** Prior research on the scalability of embedding layers has primarily focused on token embeddings, where the vocabulary is shared with the language model's decoding head. For instance, Tao et al. [2024] show that larger models benefit from larger vocabularies, reflecting trends where vocabulary sizes have grown from tens of thousands [Devlin et al., 2019, Radford et al., 2019] to hundreds of thousands of tokens [Google, 2024, Adler et al., 2024, Dubey et al., 2024, Liu et al., 2024]. However, even for the largest models, the optimal vocabulary sizes predicted by Tao et al. [2024] remain much smaller relative to model size. For instance, a vocabulary of 216K tokens for a 70B-parameter model. This relatively small token embedding layer limits the potential for expanding model capacity through token embeddings alone. To address this, Roy et al. [2022] propose decoupling input and output embeddings by introducing an additional embedding layer for bi-grams. Building on this idea, we also decouple the input and decoding embedding layers, but crucially, we parameterize the additional embedding table with a neural network during training. This approach allows us to efficiently scale the additional input embedding layer and introduces a new scaling strategy: scaling the $\mathcal{A}_{\text{f-gram}}$ model that learns to generate the additional embeddings.

Concurrently, Huang et al. [2025] propose the over-tokenized transformer, which also decouples input and decoding embeddings by introducing an additional input embedding layer for $n$-grams. They observe similar performance gains as the input embedding size increases. Unlike our approach, which selectively retains only frequent $n$-grams, they hash $n$-grams into a fixed number of embeddings to manage the vast $n$-gram space, a strategy that likely also mitigates sparse updates (Appendix D). A key distinction is that their additional embedding layer must be fully instantiated and resident on accelerators during training, leading to significant memory challenges despite tensor sharding. Moreover, through the design of $\mathcal{A}_{\text{f-gram}}$, SCONE expands model capacity not only by enlarging the embedding table but also by introducing a scalable $\mathcal{A}_{\text{f-gram}}$ model for learning the embeddings. We compare SCONE with the over-tokenized transformer in Section 4.2. Results suggest that while

both SCONE and the over-tokenized transformer show clear improvements over the baseline, SCONE offers better gains owing to the additional capacity provided by $\mathcal{A}_{\text{f-gram}}$.

**Mixture of Lookup Experts.** A concurrent work by Jie et al. [2025] proposes discretizing the inputs to a Mixture-of-Experts (MoE) layer so that, at inference time, it can be implemented as a simple lookup table. They use token embeddings from the input embedding layer as keys for the lookup. This design enables training with a larger MoE layer to improve performance, while having negligible impact on FLOPs and accelerator memory usage during inference. Their approach shares a similar insight with our design of the $\mathcal{A}_{\text{f-gram}}$ model: constructing a neural network module with discretized inputs that can be switched to an efficient lookup table at inference. However, a key difference is that we introduce a method for scaling the number of keys by constructing the set $V_{\text{f-gram}}$. This enables an additional axis of scaling, namely expanding the size of the lookup table at inference. It also allows for fully utilizing the increased model capacity during training. Without this extension, a small number of discretized keys would lead to diminishing returns when scaling the parameters of the discretized module during training.

We discuss additional related work in Appendix B due to space constraints.

# 6   Conclusion

We introduce SCONE, a scalable approach for generating $n$-gram contextualized embeddings for each input token. These embeddings are learned during training and cached in off-accelerator storage for inference. SCONE enables two new strategies for scaling language models under fixed inference-time accelerator memory and FLOPS budgets, making it particularly useful for reducing serving costs and supporting latency-sensitive applications.

We discuss limitations and promising directions for future work in Appendix A.

# Acknowledgments

The authors would like to thank Andrew Tomkins for his helpful feedback on an early draft.

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

# A Limitations and Future Work

A promising direction for future work is extending SCONE beyond short $n$-grams by enabling caching for longer queries. A central challenge lies in designing effective keys for such queries. Using raw text as keys may result in low cache hit rates, as semantically similar queries often vary at the surface level. Alternatively, using semantic embeddings as keys would require discretization techniques to map continuous embeddings into a discrete key space that supports efficient indexing.

One limitation of the current study is that we evaluate SCONE only on models with up to 3B parameters at training time. This constraint is primarily due to hardware limitations, which restrict our ability to conduct large-scale pretraining on larger models. Although the model scales we tested are widely used in real-world applications, exploring the performance of SCONE at larger scales presents an exciting direction for future research. We believe applying SCONE to larger models would be highly beneficial to the community and leave this exploration for future work. Notably, our experiments in Section 4 provide encouraging evidence, as the benefits of SCONE are consistent across all model sizes we tested.

# B Additional Related Work

**Contextualized Word Embeddings.** Words can have different meanings depending on context. Prior work has incorporated context into word embeddings, either from the entire sequence [McCann et al., 2017, Peters et al., 2018] or short $n$-grams [Gupta et al., 2019], before applying them to downstream tasks. Modern language models inherently use contextualized token embeddings, leveraging attention mechanisms. In this study, we extend the embedding layer to include contextualized f-gram embeddings for each token. A key novelty is that our approach allows embeddings to be precomputed and offloaded from accelerators, providing contextual embeddings for each token without increasing FLOPS and accelerator memory usage at inference.

**Tokenization in Language Models.** Our method assumes a predefined vocabulary from a trained tokenizer. Several popular algorithms exist for training tokenizers [Sennrich et al., 2016, Wu et al., 2016, Kudo, 2018b,a]. In this work, we use a BPE tokenizer, following prior seminal works [Radford et al., 2019, Touvron et al., 2023]. However, our method is not tied to any specific tokenization algorithm and can be applied seamlessly to others.

Tokenization-free language models have also been widely explored [Kim et al., 2016, Choe et al., 2019, Xue et al., 2022, Yu et al., 2023, Wang et al., 2024, Deiseroth et al., 2024, Meta, 2024, Pagnoni et al., 2024]. While we have not tested our method on tokenization-free models, we believe our core idea—introducing an off-accelerator embedding layer by precomputing embeddings for frequent input patterns—remains applicable.

**Mixture-of-Experts (MoE) and Memory Layers.** MoE and memory layers are two established approaches for scaling language models within a fixed FLOPS budget.

MoE layers replace traditional feedforward layers with multiple parallel "experts," activating only one or a few per token using a lightweight routing mechanism [Shazeer et al., 2017, Lepikhin et al., 2021, Fedus et al., 2022, Jiang et al., 2024, He, 2024]. This allows the model to scale by increasing the number of experts without increasing the computational cost per token. However, all experts must reside on the accelerator, resulting in significantly higher memory usage.

Memory layers, on the other hand, store large collections of embeddings (continuous vectors) and retrieve the nearest neighbors during the forward pass via (approximate) similarity search [Weston et al., 2015, Sukhbaatar et al., 2015, Lample et al., 2019, Berges et al., 2024]. These retrieved embeddings enhance the model's capabilities without adding much to the FLOPS budget. Despite improvements in similarity search techniques [Lample et al., 2019, Johnson et al., 2019], memory layers still require storing the embeddings on the accelerator, making memory demands impractically high at larger scales [Berges et al., 2024]. Furthermore, because embeddings are typically updated via backpropagation, memory layers introduce additional challenges related to sparse updates as the memory size grows.

While MoE layers, memory layers, and our proposed method SCONE all maintain fixed inference FLOPS, a key advantage of SCONE is it also maintains fixed accelerator memory usage at inference.

---

**Algorithm 2** Basic Next-Word Prediction Model $M_{\mathcal{T},\mathcal{A},\mathcal{D}}$.

---

**Parameters:**
- $\mathcal{T}: V_{\text{token}} \to \mathbb{R}^d$: token embedding layer,
- $\mathcal{A}: (\mathbb{R}^d)^{\leq T} \to \mathbb{R}^d$: transformer model, where $T$ is the maximum sequence length,
- $\mathcal{D}: \mathbb{R}^d \to \Delta_{V_{\text{token}}}$: prediction head.

**Input:** $(\sigma_1, \ldots, \sigma_m) \in V_{\text{token}}^*$ for $m \leq T$.

**Output:** Probability distribution over next token $\hat{\sigma}_{m+1}$.

**for** $i = 1, \ldots, m$ **do**

    $\boldsymbol{e}_i \leftarrow \mathcal{T}(\sigma_i)$ : Input embedding per token.

$\boldsymbol{e}_{\text{out}} \leftarrow \mathcal{A}(\boldsymbol{e}_1, \ldots, \boldsymbol{e}_m)$: Output embedding.

**return** $D(\boldsymbol{e}_{\text{out}})$

---

---

**Algorithm 3** Constructing a set of f-grams $V_{\text{f-gram}}$.

---

**Parameters:** $S$: desired size of $V_{\text{f-gram}}$.

**Input:** $\{(\sigma_1, \ldots, \sigma_T)^{(i)}\}$ : token sequences from training set.

**Output:** $V_{\text{f-gram}} \subseteq V_{\text{token}}^{[2,K]}$: set of f-grams of size $S$.

**for** $n = 2, \ldots, K$ **do**

    **for** $\omega := (\sigma_1', \ldots, \sigma_k') \in V_{\text{token}}^n$ **do**

        $C_\omega \leftarrow$ the number of times $\omega$ appears in all sequences $\{(\sigma_1, \ldots, \sigma_T)^{(i)}\}$.

Let $\omega_1, \omega_2, \ldots$ be list of elements of $\bigcup_{n=2}^K V_{\text{token}}^n$, sorted such that $C_{\omega_1} \geq C_{\omega_2} \geq \cdots$, breaking ties arbitrarily.

**return** $\{\omega_1, \ldots, \omega_S\}$: set of f-grams of size $S$.

---

By focusing on the input embedding layer, SCONE ensures that the computational overhead remains $O(1)$ during inference and enables offloading the additional parameters off-accelerator with negligible impact on end-to-end latency.

**Implicit $n$-gram Patterns in Transformers.** Recent work analyzing the internal mechanisms of transformers has shown that these models often utilize implicit $n$-gram patterns for prediction [Geva et al., 2021, 2022, Voita et al., 2024]. For instance, Chen et al. [2024] demonstrate that certain attention heads can detect specific $n$-gram patterns, while MLPs can perform linguistic operations such as adding the "-ing" suffix. These findings underscore the importance of $n$-gram information in language modeling and offer a potential explanation for the effectiveness of SCONE. An interesting future direction is to examine how SCONE's f-gram embeddings interact with the transformer's implicit $n$-gram patterns.

**Embedding Sparsity in Multilingual Applications and Recommender Systems.** This work focuses on a common setting for training LLMs: language modeling on large-scale text corpora, primarily in English. However, scaling embedding layers presents challenges beyond this context, particularly due to frequency-related performance degradation caused by sparsity. Multilingual applications are one such scenario. Two phrases in different languages may refer to the same concept but correspond to different embedding vectors. Their embeddings should ideally be close. Recent work has explored methods for learning transferable embeddings in cross-lingual settings [Artetxe et al., 2020, Chen et al., 2023]. Another relevant example is scaling the embeddings for recommender systems [Chen et al., 2019, Liu et al., 2021], where embeddings often dominate the model's parameter count due to the high cardinality of user or item categories. For both scenarios, SCONE 's strategy, i.e., parameterizing large embedding tables using a neural network, provides a complementary approach to help mitigate sparsity issues.

## C  Additional Algorithms

**Algorithm 4** Next-word prediction with SCONE $M_{\mathcal{T},V_{\text{f-gram}},\mathcal{A}_{\text{f-gram}}|\mathcal{F},\mathcal{A}_{\text{main}},\mathcal{D}}$

---

**Parameters:**

- $\mathcal{T} : V_{\text{token}} \to \mathbb{R}^d$: token embedding layer,
- $V_{\text{f-gram}} \subseteq V_{\text{token}}^{[2,K]}$: set of f-grams,
- **TRAINING**: f-gram transformer model
    - $\triangleright\ \mathcal{A}_{\text{f-gram}} : (\mathbb{R}^d)^{\leq K} \to \mathbb{R}^d$,
- **INFERENCE**: f-gram embedding layer
    - $\triangleright\ \mathcal{F} : V_{\text{f-gram}} \to \mathbb{R}^d$.
- $\mathcal{A}_{\text{main}} : (\mathbb{R}^d)^{\leq T} \to \mathbb{R}^d$: main transformer model.
- $\mathcal{D} : \mathbb{R}^d \to \Delta_{V_{\text{token}}}$: Prediction head.

**Input:** $(\sigma_1, \ldots, \sigma_m) \in V_{\text{token}}^*$ for $m \leq T$, where $T$ is the maximum sequence length.

**Output:** Probability distribution over next token $\hat{\sigma}_{m+1}$.

$(\boldsymbol{e}_1, \ldots, \boldsymbol{e}_m) \leftarrow F_{\mathcal{T},V_{\text{f-gram}},\mathcal{A}_{\text{f-gram}}|\mathcal{F}}(\sigma_1, \ldots, \sigma_m)$ (Algorithm 1)

$\boldsymbol{e}_{\text{out}} \leftarrow \mathcal{A}_{\text{main}}(\boldsymbol{e}_1, \ldots, \boldsymbol{e}_m)$

**return** $D(\boldsymbol{e}_{\text{out}})$

---

In Section 2, we discuss a simple next-word prediction model, $M_{\mathcal{T},\mathcal{A},\mathcal{D}}$, consisting of a token embedding layer $\mathcal{T}$, a transformer model $\mathcal{A}$, and a prediction head $\mathcal{D}$. This model takes a token sequence $(\sigma_1, \ldots, \sigma_m)$, with each token from the token vocabulary $V_{\text{token}}$, and produces a probability distribution for the next token. We provide the pseudocode for $M_{\mathcal{T},\mathcal{A},\mathcal{D}}$ in Algorithm 2.

In Section 3, we introduce an algorithm for constructing a set of f-grams given a target number of f-grams. We present the pseudocode for the construction process in Algorithm 3. Although Algorithm 3 clearly illustrates the procedure, it is too expensive to implement in practice. In Section 3.1, we describe an efficient implementation that requires only $K - 1$ linear scans over the pre-training corpus, where $K$ is a hyperparameter controlling the maximum length of f-grams.

In Algorithm 1 in Section 3, we present the pseudocode for SCONE's process of generating contextualized f-gram embeddings. Next, we describe the end-to-end next-word prediction process using SCONE (Algorithm 4). Specifically, the process, denoted as $M_{\mathcal{T},V_{\text{f-gram}},\mathcal{A}_{\text{f-gram}}|\mathcal{F},\mathcal{A}_{\text{main}},\mathcal{D}}$, takes an input sequence $(\sigma_1, \ldots, \sigma_m) \in V_{\text{token}}^*$ and produces a distribution over the next token $\hat{\sigma}_{m+1}$. Note that in Algorithm 4, f-gram embeddings are generated with $\mathcal{A}_{\text{f-gram}}$ during training and retrieved from a lookup table $\mathcal{F}$ during inference.

## D Challenges of Scaling Vocabulary Size in Embedding Layers

Scaling the token vocabulary size is the most straightforward way to enlarge an embedding layer, but we find that larger token vocabularies degrade performance beyond a certain threshold and significantly increase accelerator usage during decoding. We pre-train GPT-2 models [Radford et al., 2019] with three sizes of non-embedding parameters: 85M (small), 302M (medium), and 708M (large) on the WebText dataset [Peterson et al., 2019], testing six vocabulary sizes ranging from 32,768 to 2,097,152. The tokenizers are trained using the BPE algorithm [Gage, 1994, Sennrich et al., 2016]. We follow the implementation in Tao et al. [2024], which allows token merges across word boundaries. Each model is trained on 80B tokens. Since larger vocabularies produce fewer tokens for the same dataset, they effectively enable models to process more data. Additional implementation details such as training hyperparameters are provided in Appendix F.1.

Figure 8 presents the average bits per character (BPC) on the WebText validation set. We report BPC instead of cross-entropy loss because the latter is sensitive to vocabulary size, with larger vocabularies typically producing higher losses. BPC, by contrast, is a common vocabulary-insensitive metric for comparing models trained with different tokenizers [Huang et al., 2024]. We observe that BPC for all three models initially improves with larger vocabulary sizes but eventually deteriorates.

Figure 9 shows the percentages of tokens that receive more than a given number of updates over 100M training tokens. In standard embedding layers, gradients are directly backpropagated to the embedding vectors. With a fixed number of training tokens, larger vocabularies lead to fewer updates per token. For a vocabulary size of 2,097,152, only 7.3% of tokens receive more than 100 updates,

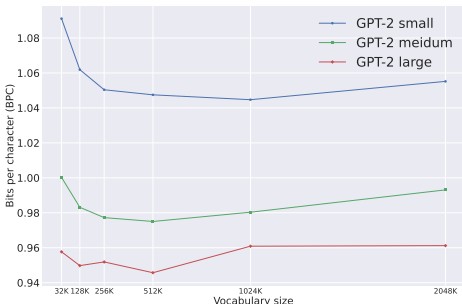

Figure 8: BPC of three model sizes on the validation set (lower is better). For all three model sizes, BPC initially improves as vocabulary size increases but eventually deteriorates.

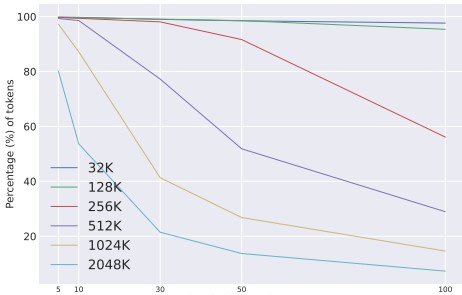

Figure 9: Percentages of tokens ($y$-axis) that receive more than a given number of updates ($x$-axis), measured over 100M training tokens. As the vocabulary size increases, tokens receive increasingly sparse updates.

compared to 97.6% for a vocabulary size of 32,768. This suggests that the performance drop for larger vocabularies may stem from sparse updates to per-token embedding vectors.

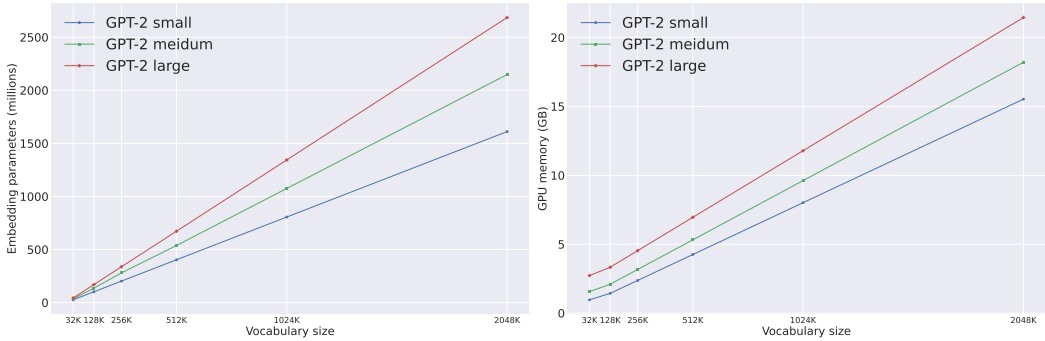

Figure 10: Number of embedding layer parameters stored on GPU and the corresponding memory usage. For large vocabulary sizes, the memory usage increases linearly with the vocabulary size.

In addition to performance degradation, increasing the vocabulary size significantly raises accelerator usage during the inference stage. This is because predicting the next token involves running a linear layer and softmax operation across the entire vocabulary to identify the closest embedding. Figure 10 illustrates that both the number of embedding layer parameters stored on the GPU and the GPU memory cost increase linearly with vocabulary size. These costs are measured using a batch size of 1, a sequence length of 1024, and 16-bit precision.

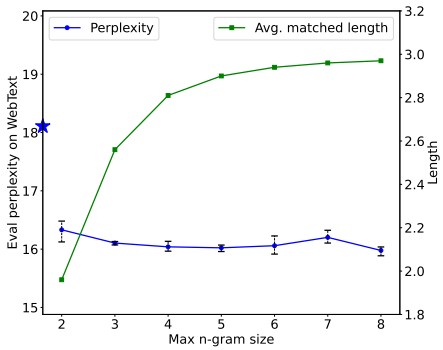

Figure 11: Effect of the maximum f-gram length in $V_{\text{f-gram}}$, evaluated on the WebText validation split.

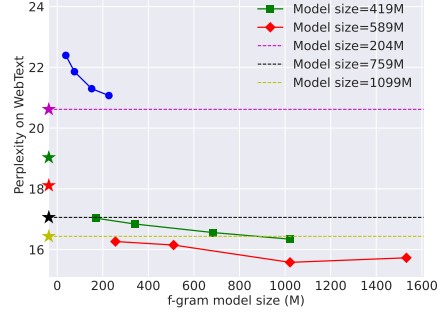

Figure 12: Evaluation perplexity on WebText as a function of the size of $\mathcal{A}_{\text{f-gram}}$.

# E    Additional Experiments

## E.1    More Results for Training on WebText

**Varying Maximum f-gram Length.**    In Section 4.1.1, we discuss the impact of varying the maximum f-gram length in $V_{\text{f-gram}}$ and present results on Wikitext-103. We observe that a relatively small maximum length is sufficient, as long as it is not too small, otherwise, the number of available $n$-grams for ranking becomes too limited. Here, in Figure 11, we show the corresponding results on WebText, which exhibit similar trends. The left $y$-axis represents the evaluation loss (averaged over three seeds), with the leftmost star indicating baseline performance. The right $y$-axis shows the average length of matched f-grams. As the maximum size increases, the loss initially decreases but then plateaus with some fluctuations. Meanwhile, the matched length rises initially before stabilizing for larger values.

**Varying $\mathcal{A}_{\text{f-gram}}$ Model Size.**    In Section 4.1.3, we discuss the impact of varying the size of $\mathcal{A}_{\text{f-gram}}$ on evaluation perplexity for Wikitext-103. We find that increasing the model size leads to further performance improvements for a fixed $|V_{\text{f-gram}}|$. In Figure 12, we present the results on WebText, which show a similar trend. Model sizes in the legend correspond to inference-time sizes on accelerators. Dashed lines and stars on the left represent baseline performance. The evaluation perplexity improves as the size of $\mathcal{A}_{\text{f-gram}}$ grows.

## E.2    Additional Results for Training on the OLMo Corpus

We present training curves and perplexity evaluations for models trained on the OLMo corpus. For SCONE, we introduce an additional $\mathcal{A}_{\text{f-gram}}$ model size of 0.6B, alongside the 1.8B $\mathcal{A}_{\text{f-gram}}$ model discussed in Section 4.2. Due to the large number of experiments, all models in this section are trained for 200B tokens, constrained by computational resources. This differs from Section 4.2, where SCONE-enabled models are trained for 500B tokens and baseline models for 1T tokens.

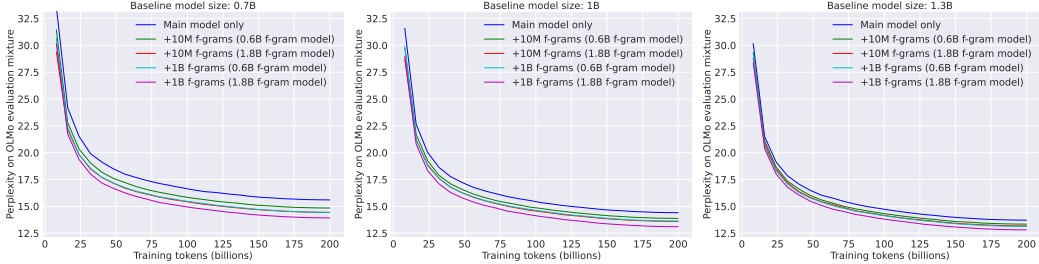

Figure 13: Average perplexity on the OLMo evaluation mixture throughout training. Models with SCONE enabled converge later, indicating stronger capacity, and achieve better perplexity.

**Training Curves.** Figure 13 shows the evaluation perplexity curves for OLMo-0.7B, OLMo-1B, and OLMo-1.3B throughout training. The curves indicate that models trained with SCONE converge more slowly, suggesting that SCONE effectively increases model capacity. Furthermore, both increasing the number of f-grams and enlarging the $\mathcal{A}_{\text{f-gram}}$ model size enhance model capacity.

Table 3: SCONE consistently improves perplexity (lower is better) across all evaluation corpora. We train three baseline models with sizes of 1B, 1.3B, and 1.9B parameters. For the 1B and 1.3B baseline models, we apply SCONE using four different configurations and present the results directly below each corresponding baseline.

| Model size | c4-en | books | common-crawl | pes2o | reddit | stack | wiki | ice | m2de-s2orc | pile | wikitext-103 | Average |
|---|---|---|---|---|---|---|---|---|---|---|---|---|
| **1B baseline** | 16.813 | 21.570 | 16.752 | 11.682 | 22.612 | 3.360 | 14.453 | 15.281 | 27.900 | 10.429 | 16.053 | 16.082 |
| +10M $V_{\text{f-gram}}$ (0.6B $\mathcal{A}_{\text{f-gram}}$) | 16.087 | 20.963 | 16.039 | 11.270 | 21.797 | 3.274 | 13.777 | 14.979 | 26.361 | 10.128 | 15.371 | 15.459 |
| +10M $V_{\text{f-gram}}$ (1.8B $\mathcal{A}_{\text{f-gram}}$) | 15.727 | 20.429 | 15.473 | 11.124 | 21.388 | 3.231 | 13.454 | 14.709 | 25.785 | 9.956 | 15.104 | 15.125 |
| +1B $V_{\text{f-gram}}$ (0.6B $\mathcal{A}_{\text{f-gram}}$) | 15.846 | 20.593 | 15.684 | 11.071 | 21.411 | 3.213 | 13.543 | 14.702 | 26.026 | 9.889 | 15.077 | 15.187 |
| +1B $V_{\text{f-gram}}$ (1.8B $\mathcal{A}_{\text{f-gram}}$) | 15.158 | 19.680 | 14.857 | 10.761 | 20.757 | 3.133 | 12.964 | 14.220 | 24.958 | 9.553 | 14.354 | 14.581 |
| **1.3B baseline** | 15.994 | 20.157 | 15.921 | 11.148 | 21.634 | 3.248 | 13.721 | 14.651 | 26.583 | 9.927 | 15.143 | 15.284 |
| +10M $V_{\text{f-gram}}$ (0.6B $\mathcal{A}_{\text{f-gram}}$) | 15.509 | 19.816 | 15.407 | 10.887 | 21.022 | 3.192 | 13.260 | 14.372 | 25.450 | 9.757 | 14.616 | 14.844 |
| +10M $V_{\text{f-gram}}$ (1.8B $\mathcal{A}_{\text{f-gram}}$) | 15.193 | 19.587 | 14.995 | 10.795 | 20.735 | 3.171 | 13.071 | 14.272 | 25.258 | 9.674 | 14.438 | 14.654 |
| +1B $V_{\text{f-gram}}$ (0.6B $\mathcal{A}_{\text{f-gram}}$) | 15.270 | 19.510 | 15.106 | 10.707 | 20.763 | 3.139 | 13.073 | 14.177 | 25.009 | 9.546 | 14.397 | 14.609 |
| +1B $V_{\text{f-gram}}$ (1.8B $\mathcal{A}_{\text{f-gram}}$) | 14.803 | 18.996 | 14.541 | 10.502 | 20.296 | 3.085 | 12.637 | 13.971 | 24.533 | 9.357 | 13.971 | 14.245 |
| **1.9B baseline** | 15.270 | 19.017 | 15.184 | 10.719 | 20.752 | 3.163 | 13.119 | 14.095 | 25.461 | 9.570 | 14.229 | 14.598 |

**Perplexity Evaluation.** Figure 2 presents perplexity results on the OLMo evaluation mixture[3], which covers 11 diverse corpora, including web crawl data, literature, online forums, scientific writing, coding, and more. Table 3 details the performance breakdown by corpus. Results show that increasing both $|V_{\text{f-gram}}|$ and the size of $\mathcal{A}_{\text{f-gram}}$ consistently improves performance across all datasets.

Additionally, Figure 2 reports token generation speeds measured using the vLLM framework [Kwon et al., 2023] with a batch size of 1. Even with large $|V_{\text{f-gram}}|$, embedding retrieval remains efficient and is not a bottleneck for inference.

As a representative example, in the 1B model variant, the baseline achieves an average perplexity of 16.082. Setting $|V_{\text{f-gram}}|$ to 10M improves perplexity to 15.459 with a 0.6B $\mathcal{A}_{\text{f-gram}}$ model and to 15.125 with a 1.8B $\mathcal{A}_{\text{f-gram}}$ model—the latter outperforming the 1.3B baseline (15.284). Further increasing $|V_{\text{f-gram}}|$ to 1B improves perplexity to 15.187 (0.6B $\mathcal{A}_{\text{f-gram}}$) and 14.581 (1.8B $\mathcal{A}_{\text{f-gram}}$), surpassing the 1.9B baseline (14.598) while requiring only about half the FLOPS and accelerator memory at inference time.

### E.3 Apply SCONE in Post-training

We apply SCONE to supervised fine-tuning of Qwen3-4B-base using the open-r1[4] framework and the open-r1/Mixture-of-Thoughts dataset[5]. In this setup, Qwen3-4B serves as the main model, while Qwen3-8B-base or Qwen3-14B-base are used as the f-gram models. We set the number of f-grams to 10M and follow the training hyperparameters in open-r1. Table 4 compares the resulting SCONE-enabled models with the Qwen3-4B baseline in terms of both accuracy and decoding latency.

| Model | AIME 2024 pass@1 | LiveCodeBench v4_v5 pass@1 | Decoding Latency (per-token) (ms) |
|---|---|---|---|
| Qwen3-4B-base | 45.3 | 30.8 | 10.05 |
| SCONE-4B (8B f-gram model) | 48.3 (+3.0) | 34.5 (+3.7) | 10.13 |
| SCONE-4B (14B f-gram model) | 51.6 (+6.3) | 36.3 (+5.5) | 10.13 |

Table 4: Performance comparison of Qwen3-4B variants on AIME 2024 and LiveCodeBench v4/v5 with decoding latency.

---

[3]`https://github.com/allenai/OLMo/blob/v0.4.0/configs/official/OLMo-1B.yaml#L90`
[4]`https://github.com/huggingface/open-r1`
[5]`https://huggingface.co/datasets/open-r1/Mixture-of-Thoughts`

These results show that SCONE consistently improves performance over the baseline, with larger f-gram models yielding greater gains, all while maintaining similar inference latency.

### E.4 Summary of Comparison on Computational Resources

In Table 5, we summarize the key metrics of SCONE for the three model variants evaluated in Section 4.2. The metrics include (1) GPU memory, (2) CPU memory, (3) disk usage, and (4) FLOPS/latency for training and inference. All measurements are taken with a context length of 2048 and a batch size of 4 on a single A100 80 GB GPU. The three settings are: (1) the 1.9B baseline model, (2) SCONE 1.3B, with a 1.8B f-gram model and 10M cached f-gram embeddings, and (3) SCONE 1B , with a 1.8B f-gram model and 1B cached f-gram embeddings.

| Model Variant | Peak GPU Memory (GB) | CPU Memory Overhead (GB) | Disk Usage (TB) | Decoding Latency (per-token) (ms) | Training FLOPS (per-seq.) ($\times 10^{13}$) |
|---|---|---|---|---|---|
| 1.9B baseline | 8.38 | N/A | N/A | 6.45 | 2.73 |
| SCONE-1.3B (10M f-grams) | 5.60 | 41.76 | N/A | 4.83 | 4.94 |
| SCONE-1B (1B f-grams) | 4.45 | N/A | 7.67 | 4.90 | 5.57 |

Table 5: Comparison of SCONE model variants on memory, storage, latency, and compute efficiency.

Both memory and disk usage are reported for inference. The decoding latency is averaged over one thousand decoding steps. As shown in Section 4.2, all three models achieve similar downstream performance. Compared to the 1.9B baseline, SCONE-enabled models significantly reduce GPU memory usage and decoding latency, at the cost of increased CPU memory or disk storage.

## F  Implementation Details

We provide additional implementation details below. Most of our experiments are conducted on 4 8×H100 nodes, while some experiments are conducted on 2 16×A100 nodes.

While f-gram lookup is efficient for inference, it creates a bottleneck during training since at training time transformer models process all token positions in parallel. This leads to GPU idle time when fetching the longest matching f-gram on the fly. To remove this bottleneck, after we construct the set of f-grams ($V_{\text{f-gram}}$), we pre-scan the training sequences to tag the longest matching length for each token. During training, we can then directly retrieve the corresponding f-gram for forward computation with the $\mathcal{A}_{\text{f-gram}}$ model.

For the $\mathcal{A}_{\text{f-gram}}$ model, we use an absolute position embedding layer where the maximum position equals the longest $n$-gram in $V_{\text{f-gram}}$. Within each batch, all f-grams are padded to the longest $n$-gram length in that batch. We train all models with the bfloat16 precision.

### F.1  WebText

For pre-training on WebText [Peterson et al., 2019], we follow Radford et al. [2019] and set the batch size and sequence length to 512 and 1024, respectively. Radford et al. [2019] do not specify the number of training tokens or optimizer details. We train the models for 80B tokens, roughly doubling the count in Radford et al. [2018]. For optimization, we use AdamW [Loshchilov and Hutter, 2019] with a weight decay of 0.1. Following Hoffmann et al. [2022], we set the maximum learning rate to $2 \times 10^{-4}$ and apply a cosine learning rate scheduler. We list the model configurations in Table 6.

| Parameters (M) | d_model | ffw_size | n_layers |
|---|---|---|---|
| 128 | 1024 | 4096 | 6 |
| 204 | 1024 | 4096 | 12 |
| 491 | 1536 | 6144 | 12 |
| 759 | 1536 | 6144 | 24 |
| 589 | 1536 | 6144 | 18 |
| 1099 | 1536 | 6144 | 36 |

Table 6: Baseline model configurations for pre-training on WebText. For constructing the f-gram model ($\mathcal{A}_{\text{f-gram}}$), we vary the number of layers in the 128M, 491M, and 589M variants and discard the token embedding layer.

| Parameters (M) | d_model | ffw_size | n_layers |
|---|---|---|---|
| 711 | 2048 | 8192 | 12 |
| 1014 | 2048 | 8192 | 18 |
| 1316 | 2048 | 8192 | 24 |
| 1920 | 2048 | 8192 | 36 |

Table 7: Model configurations for pre-training on the OLMo corpus. To construct the f-gram model ($\mathcal{A}_{\text{f-gram}}$), we use the 711M and 1920M variants, excluding the token embedding layers.

## F.2    OLMo Tokenized Training Corpus

For pre-training on the OLMo tokenized training corpus, we follow the optimizer settings for the 1B variant in Groeneveld et al. [2024] [6]. All models use a sequence length of 2048. We use DeepSpeed [DeepSpeed, 2024] with ZeRO stage 1 that partitions the optimizer state across GPUs to reduce GPU memory usage. We list the model configurations in Table 7.

---

[6]`https://github.com/allenai/OLMo/blob/v0.4.0/configs/official/OLMo-1B.yaml#L40`

