# OpenReview forum: "Scaling Embedding Layers in Language Models"
_NeurIPS.cc/2025/Conference — NeurIPS 2025 poster_

### Official Review · Reviewer_H6nF · 2025-06-20

**Clarity:** 4
**Significance:** 2
**Originality:** 3
**Rating:** 5
**Confidence:** 5

**Summary:**

This paper proposes enhancing LLM performance by incorporating n-gram embeddings. To control the size of the n-gram embeddings, the authors suggest building embeddings only for high-frequency n-grams. They demonstrate that scaling the size of the n-gram model and increasing the number of n-grams both effectively reduce the perplexity of LLMs. A key advantage of this approach is that n-gram embeddings are context-free, allowing them to be read directly without recomputation. However, a major drawback is the substantial storage overhead—since the total number of global n-grams scales with the fourth power of the vocabulary size, the method can only accommodate a limited number of n-grams in practice.

**Questions:**

see cons.

**Ethical Concerns:**

["NO or VERY MINOR ethics concerns only"]

**Final Justification:**

I think after adding the rebuttal experiments, the paper should be accepted.

**Limitations:**

Yes.

**Quality:**

3

**Strengths And Weaknesses:**

**Pros**:

No recomputation required: The n-gram embeddings are context-free, meaning they can be directly retrieved without recomputation, unlike context-dependent representations. This leads to low inference overhead.

Effective performance improvement: Experiments show that increasing both the number and dimensionality of n-gram embeddings significantly reduces the perplexity (PPL) of LLMs, indicating the method effectively enhances language modeling capability.

Clear writing and logic: The paper is well-written with clear logic, making it easy to read and understand.

**Cons**:

Large storage overhead: The storage cost is substantial. Loading the n-gram embeddings during the prefill phase incurs significant overhead, although the cost is more manageable during decoding.

Not a game changer: The improvement in perplexity, while present, is relatively limited and may not be sufficient to position the method as a major advancement.

Limited coverage: It is not feasible to compute embeddings for all n-grams; only a limited number can be supported. As a result, the method may offer limited benefits for low-frequency or rare tasks.

Lack of architectural generalization: The method's generality across architectures is not explored. Demonstrating its effectiveness on alternative models such as Mamba or GLA could significantly strengthen the paper's claims.

---

> ### Author Rebuttal · Authors · 2025-07-30
>
> Thank you for the review and comments! Please see our response below.
>
> **1. Loading the n-gram embeddings during the prefill phase incurs significant overhead, although the cost is more manageable during decoding.**
>
> We thank the reviewer for raising this point. We agree that during the prefill phase, substantially more n‑gram embeddings must be accessed compared to the decoding phase, particularly for long-context tasks. However, techniques such as chunked prefill are commonly used to improve efficiency. This is because processing a few hundred (or even just a few dozen) tokens in parallel is typically enough to fully saturate the accelerator. Under these conditions, SCONE can load hundreds of token embeddings without adding noticeable latency beyond what is shown in Figure 7. When the n‑gram embedding table is stored in CPU memory, this loading can be handled efficiently with batched operations. If the embedding tables are stored on disk, the solution is to increase the number of parallel read processes. We will incorporate this discussion in the revision.
>
>
> **2. The improvement in perplexity, while present, is relatively limited.**
>
> In addition to the perplexity evaluation, our submission also reports downstream task results in Table 1 (Section 4.2). These results show that the best SCONE‑1B variant outperforms a 1.9B baseline across multiple tasks. While the improvements are apparently modest, we believe they are nonetheless meaningful and demonstrate the effectiveness of our approach.
>
>
> **3. Only a limited number of n-grams can be supported. As a result, the method may offer limited benefits for low-frequency or rare tasks.**
>
> While the zero-shot evaluation results in Section 4.2 (covering math, coding, factuality QA, and other tasks) show that the f-gram set built from a diverse pre-training corpus can generalize well across a broad range of tasks, we acknowledge that f-gram coverage may be limited for extremely rare corpora.
>
> To partially address this concern, as discussed in our response to Reviewer 5vFb, we show that SCONE can also be applied at the **post-training** stage. This allows the **f-gram embedding table to be updated on a new corpus without having to repeat the entire pre-training process**.
>
> Specifically, we applied SCONE to the supervised fine-tuning (SFT) of [Qwen3-4B-base](https://huggingface.co/Qwen/Qwen3-4B-Base) using [open-r1](https://github.com/huggingface/open-r1) and the [open-r1/Mixture-of-Thoughts](https://huggingface.co/datasets/open-r1/Mixture-of-Thoughts) dataset. In our setup, Qwen3-4B serves as the main model, and Qwen3-8B-base or Qwen3-14B-base are used as the f-gram models. We set the number of f-grams to 10M and follow the training hyperparameters in open-r1. The table below compares the resulting SCONE-enhanced models with the Qwen3-4B baseline in terms of both accuracy and decoding latency.
>
> |   |  AIME 2024 pass@1  |  LiveCodeBench v4_v5 pass@1 | Decoding Latency at inference (per-token) |
> |----------|----------|----------|----------|
> | Qwen3-4B-base Baseline| 45.3 | 30.8 | 10.05 ms |
> |Qwen3-4B SCONE (Qwen3-8B f-gram model) | 48.3 (+3.0) | 34.5 (+3.7) | 10.13 ms |
> |Qwen3-4B SCONE (Qwen3-14B f-gram model) | 51.6 (+6.3)  | 36.3 (+5.5)  | 10.13 ms |
>
> These results indicate that both the f-grams and the f-gram embedding table can be updated during post-training, without requiring a full re-run of pre-training.
>
> **4. The method's generality across architectures is not explored.**
>
> While the transformer architecture we used is a common choice in the literature, we agree that evaluating SCONE on alternative architectures such as Mamba or GLA would be valuable. Unfortunately, due to limitations in computational resources and human effort, such an extensive study is beyond the scope of the current work. In the revision, we will include this point in the Limitations section and pose it as an interesting research direction to pursue.

---

### Official Review · Reviewer_5vFb · 2025-06-24

**Clarity:** 3
**Significance:** 3
**Originality:** 3
**Rating:** 4
**Confidence:** 4

**Summary:**

This work introduced SCONE, which extends input embedding layers to decrease the decoding costs. SCONE adds additional embeddings for a set of frequent n-grams. These newly added embeddings are learned with a separate model during training. During inference, these embeddings are pre-computed and stored in memory. The experiments demonstrate	the scaling law with regards to the number of n-gram embeddings and the size of the model that used to learn n-gram embeddings.

**Questions:**

See Strengths And Weaknesses

**Ethical Concerns:**

["NO or VERY MINOR ethics concerns only"]

**Final Justification:**

The authors' rebuttal has addressed some of my concerns, in terms of including math and code tasks. I would like to raise my score to 4.

**Limitations:**

Yes

**Quality:**

3

**Strengths And Weaknesses:**

Strength:
1. The paper is well-written and presents detailed results for scaling experiments.


Weaknesses:
1. Lack of experiments for larger model. The scaling experiments are mainly conducted with 76M/340M/510M parameters, which is limited in scale. How about the performance of SCONE applied for larger model, e.g., Qwen2.5-3B/7B?
2. The evaluation on downstream tasks only focus on language modeling tasks. Including more non-generation tasks, e.g., math or code tasks, could enhance the effectiveness of the proposed method, since these tasks usually require the model to generate very long context.
3. The proposed method seems quit similar to a variant of prefix tuning. Could the authors discuss the difference between prefix tuning and SCONE? Is it possible to use SCONE for post-training stage of pre-trained LLM?

---

> ### Author Rebuttal · Authors · 2025-07-30
>
> We thank the reviewer for their review and suggestions. Following the reviewer’s suggestion, we have added new results applying SCONE during the post-training stage on larger models. Please find our detailed response below.
>
> **1. The scaling experiments are mainly conducted with 76M/340M/510M parameters, which is limited in scale. How about the performance of SCONE applied for larger model, e.g., Qwen2.5-3B/7B?**
>
> While the scaling experiments in Section 4.1 are based on GPT-2 sized models, we also include experiments with OLMo-1B in Section 4.2, which is a commonly used efficient language model, and compare against baselines up to 1.9B parameters. As we acknowledge in the Limitations section (Appendix A), pre-training larger models requires substantial computational resources and is beyond the scope of this work.
>
> Nevertheless, as suggested in the third question, we conducted additional post-training experiments on the latest Qwen3 models, since post-training is much less resource-intensive than pre-training. The results of these experiments can be found in our response to item 4 below. We hope these additional results can showcase the effectiveness of SCONE in larger-model settings.
>
> **2. Including more non-generation tasks, e.g., math or code tasks, could enhance the effectiveness of the proposed method, since these tasks usually require the model to generate very long context.**
>
> In Section 4.2 of our submission, we report the performance on the MMLU tests used in the OLMo repo, which includes math and coding related problems. In our new experiments for post-training with Qwen3 models, we also report results on two additional benchmarks: AIME (math) and LiveCodeBench (code).
>
> **3. Could the authors discuss the difference between prefix tuning and SCONE?**
>
> [Prefix tuning](https://arxiv.org/abs/2101.00190) is a parameter-efficient fine-tuning approach that optimizes a small set of continuous vectors prepended to the model’s input context. While both SCONE and prefix tuning involve learning additional embeddings for a language model, SCONE differs in two key ways: (1) SCONE learns embeddings for a predefined set of n-grams, which can be cached and reused at inference time; and (2) SCONE generates these embeddings using a separate model, which can be scaled up during training to improve performance. Importantly, as the n-gram embeddings can be cached, scaling this separate model does not increase accelerator usage at inference.  We will highlight these points better in the revision.
>
>
> **4. Is it possible to use SCONE for the post-training stage of pre-trained LLM?**
>
> Yes.  We conducted additional experiments to demonstrate that SCONE can indeed be applied during the post-training stage. Specifically, we use SCONE with supervised fine-tuning (SFT) to post-train [Qwen3-4B-base](https://huggingface.co/Qwen/Qwen3-4B-Base), following the [open-r1](https://github.com/huggingface/open-r1) framework. The SFT dataset is [open-r1/Mixture-of-Thoughts](https://huggingface.co/datasets/open-r1/Mixture-of-Thoughts). In our setup, Qwen3-4B serves as the main model, while Qwen3-8B-base or Qwen3-14B-base are used as the f-gram models. We set the number of f-grams to 10M and follow the same training hyperparameters as open-r1. The table below presents the new results, comparing our SCONE-based model with the Qwen3-4B baseline in terms of both performance and latency.
>
>
> |   |  AIME 2024 pass@1  |  LiveCodeBench v4_v5 pass@1 | Decoding Latency at inference (per-token) |
> |----------|----------|----------|----------|
> | Qwen3-4B-base Baseline| 45.3 | 30.8 | 10.05 ms |
> |Qwen3-4B SCONE (Qwen3-8B f-gram model) | 48.3 (+3.0) | 34.5 (+3.7) | 10.13 ms |
> |Qwen3-4B SCONE (Qwen3-14B f-gram model) | 51.6 (+6.3)  | 36.3 (+5.5)  | 10.13 ms |
>
> These results show that incorporating 8B and 14B f-gram models yields clear improvements on both AIME and LiveCodeBench, with larger f-gram models providing greater gains. Importantly, scaling up the f-gram model does not increase inference latency because the f-gram embeddings are pre-computed and cached.
>
> Thank you for suggesting the post-training direction (and suggesting to use Qwen).  We will include these new results in the revision.

---

> > ### Comment · Reviewer_5vFb · 2025-08-04
> >
> > Thanks for the authors' additional experiments on Qwen3 models. Could the authors provide more inference speedup of Qwen3-4B SCONE (Qwen3-8/14B f-gram model) compared to the baseline?

---

> ### Comment · Reviewer_5vFb · 2025-08-05
>
> Thanks for the additional clarifications. The authors' rebuttal has addressed some of my concerns. I would like to raise my score.

---

> > ### Author Response · Authors · 2025-08-05
> >
> > Thank you again for your thoughtful suggestions and constructive feedback!

---

### Official Review · Reviewer_yxCy · 2025-06-30

**Clarity:** 4
**Significance:** 4
**Originality:** 4
**Rating:** 5
**Confidence:** 3

**Summary:**

This paper focuses on scaling embedding tables for large language models (LLMs). The authors propose a solution that (1) enables effective scaling of embedding tables and (2) is reasonably practical for both training and inference. Experimental results are strong, demonstrating improved performance when trained with equivalent resources and served using fewer resources.

**Questions:**

**Main Questions:**

1. Regarding Figure 7, could you provide specific latency metrics such as p50, p90, and p99? Usually I imagine for embedding table lookup, training is fine since you can do pipelining but it is on the critical path for inference. See also the paper I mention below.
2. Can you share e2e latency numbers for Figure 7?

**Clarifying Questions:**

1. What are the dimensions of the embedding tables?
2. I am not familiar with the Lightning Memory-Mapped Database. Does it utilize CPU memory for caching? Do you have any data on cache miss behavior?
3. On page 8, you mention that on-disk storage incurs an 86% memory overhead. Do you have insights into the reasons behind this?

**Minor Suggestions / Nits:**

1. I understand why you don't want to offload the "embedding table" for f-grams in training since you don't want to materialize it and you still want to write to it. But that reminds me of "Toward 100TB Recommendation Models with Embedding Offloading" which handles it by cache (roughly 10% of total table) and pipelining.
2. Page 3 "During training, it is parameterized by ..." this sentence is pretty confusing to me.
3. Page 5 "... instantiate a full embedding table during training" would be much better if you add this is the f-gram table.

**Ethical Concerns:**

["NO or VERY MINOR ethics concerns only"]

**Final Justification:**

Same score.

**Quality:**

4

**Strengths And Weaknesses:**

**Strengths:**

* The paper is extremely well-written, with clear motivation and thorough explanations that proactively address potential questions.
* The literature review is comprehensive.
* The ideas presented are natural, and the design choices are well-justified.
* The solution is co-designed for both training and inference.

**Weaknesses:**

* The main tradeoffs (GPU memory, disk usage, latency) for training and inference are scattered throughout the paper. The authors should consider summarizing key metrics in a table—potentially in the Appendix—including (1) GPU memory, (2) CPU memory, (3) disk usage, and (4) latency for both training and inference.
* The sell of the paper is "more training resources usage but equal inference resources usage for better performance", with the assumption that (1) inference usage is way more than training resources and (2) allocating additional resources to training in this manner is the most effective way to improve performance. While I agree with (1) and somewhat (2), it remains to see if this method is truly the best way to spend the extra training resources.

---

> ### Author Rebuttal · Authors · 2025-07-30
>
> Thank you for your review and suggestions! We have incorporated them to improve our manuscript. Please find our responses below.
>
> **1. The authors should consider summarizing key metrics in a table—potentially in the Appendix—including (1) GPU memory, (2) CPU memory, (3) disk usage, and (4) latency for both training and inference.**
>
> Thank you for the suggestion.  The following table summarizes the computational resource usage for the three model variants evaluated in Section 4.2. All measurements are taken with a context length of 2048 and a batch size of 4 on a single A100 80 GB GPU. The three settings are:
> (1) the 1.9B baseline model,
> (2) SCONE 1.3B, with a 1.8B f‑gram model and 10M cached f‑gram embeddings, and
> (3) SCONE 1B , with a 1.8B f‑gram model and 1B cached f‑gram embeddings.
>
> | Model Variant | Peak GPU Memory   |  CPU Memory Overhead | Disk Usage | Decoding latency (per-token)  | Training FLOPS (per-seq.) |
> |----------|----------|----------|----------|----------|----------|
> | 1.9B baseline |  8.38 GB | N/A | N/A | 6.45 ms | $2.73\times 10^{13}$ |
> | SCONE 1.3B (10M f-grams) | 5.60 GB  | 41.76 GB  | N/A | 4.83 ms  | $4.94\times 10^{13}$ |
> | SCONE 1B (1B f-grams) | 4.45 GB | N/A | 7.67 TB | 4.90 ms   | $5.57\times 10^{13}$ |
>
> Both memory and disk usage are reported for inference. The decoding latency is averaged over one thousand decoding steps. As shown in Section 4.2, all three models achieve similar downstream performance. Compared to the 1.9B baseline, SCONE-enabled models significantly reduce GPU memory usage and decoding latency, at the cost of increased CPU memory or disk storage.  As you suggest, we will add this summary table in the revision.
>
>
> **2. … with the assumption that (1) inference usage is way more than training resources and (2) allocating additional resources to training in this manner is the most effective way to improve performance. While I agree with (1) and somewhat (2), it remains to see if this method is truly the best way to spend the extra training resources.**
>
> While in Section 4.2 we keep the total training FLOPs for SCONE roughly the same as for the 1.9B baseline to show that SCONE provides a better use of training resources, we acknowledge that our experiments do not yet explore broader model families, such as significantly larger models. Therefore, whether this approach is always the most effective way to allocate additional training resources remains an open question and is an interesting direction for future work. We will include a discussion of this point in the ‘Limitations and Future Work’ section.
>
> That said, as discussed in Section 1, there are important scenarios where training resources could be relatively abundant but inference resources are strictly constrained. SCONE is particularly well-suited for such settings.  Two such scenarios are 1) deploying models on edge devices with limited compute, and 2) latency-sensitive applications where inference FLOPs cannot exceed some threshold.
>
> **3. Regarding Figure 7, could you provide specific latency metrics such as p50, p90, and p99?**
>
> We summarize the specific latency matrics in the table below.
>
> **Setup.** We report the largest configurations from Fig. 7: 1) 100 M f‑gram embeddings held in RAM, 2)  1 B f‑gram embeddings stored on NVMe.
> The numbers are per‑token latencies averaged over 100, 000 batches and are expressed in milliseconds (ms). For batch sizes larger than 1, the total batch time is divided by the number of tokens, i.e., the latency is amortized over all tokens in the same batch.
>
> |   |  Batchsize = 1  |  2 | 4 | 8 | 16 |
> |----------|----------|----------|----------|----------|----------|
> | p50 (RAM)|  0.016 | 0.058 | 0.046 | 0.027 | 0.017 |
> | p90  (RAM)| 0.021 | 0.081 | 0.065 | 0.034 | 0.021 |
> | p99 (RAM) | 0.026 | 0.107 | 0.076 | 0.050 | 0.032 |
> | p50 (disk) | 0.783  | 1.105 | 1.158 | 1.029 |0.783 |
> | p90 (disk) | 5.438 | 3.094 | 2.288 | 1.670 | 1.214 |
> | p99 (disk) |  6.675 | 3.946 | 2.968 | 2.211 | 1.522 |
>
> **Observations.**
>  - In‑memory (100 M). The distribution is tight: p50 essentially matches the mean reported in Fig. 7, and p99 is within 2 $\times$ p50.
>
> - On-disk (1B). Variance is higher: p99 can be up to  $\sim$ 8.5  $\times$ p50 for batchsize = 1. Increasing the batch size amortizes the look‑up cost across tokens, shrinking the p99/p50 ratio to  $\sim$ 2.8 $\times$ at batch = 16.
>
> In the revision, we will extend Figure 7 with a percentile plot (p50 / p90 / p99) numbers in the Appendix.
>
> **4. Can you share e2e latency numbers for Figure 7?**
>
> In the submission, we report the end-to-end decoding speed in the right panel of Figure 2. As shown in our response to the first question, a 1B model with a 1B f-gram embedding table stored on an NVMe drive achieves a per-token decoding latency of 4.90 ms (approximately 200 tokens per second). This is still significantly faster than the 1.9B baseline, which has an average latency of 6.45 ms, while delivering comparable downstream performance.
>
>
> **5. Clarifying Questions**
>
> **5.1** What are the dimensions of the embedding tables?
>
> The default embedding dimension is 2048, except in Section 4.1, where we conduct ablation experiments using GPT with a smaller embedding dimension of 1024.
>
> **5.2** Does Lightning Memory-Mapped Database utilize CPU memory for caching? Do you have any data on cache miss behavior?
>
> LMDB does not add an extra cache. The database file is memory‑mapped as every page on disk is assigned a position in the process’s virtual memory address (on a 64‑bit process the space is vast enough to cover terabytes). The memory mapping process itself does not load any data into RAM yet. As LMDB code touches a page, the CPU triggers a page fault, the kernel pulls that page from disk into the OS page cache, and execution continues as if the page had always been in memory.
> Because our embedding database is many times larger than the machine’s physical RAM, most look‑ups require pages that are not in the cache, so the latencies we report largely reflect disk‑bound access.
>
> **5.3** On page 8, you mention that on-disk storage incurs an 86% memory overhead. Do you have insights into the reasons behind this?
>
> The ~86% larger on‑disk footprint is primarily due to internal fragmentation from fixed‑size pages. Unlike the contiguous in‑memory layout, the on‑disk format is partitioned into fixed blocks, many of which are only partially filled: especially under small, frequent updates. Beyond that, each page carries metadata (e.g., headers, checksums). These factors collectively account for most of the observed overhead.  We are happy to add this description in the revision.
>
>
> **6. Suggestions**
>
> **6.1** Materialize the f-gram embedding table at training and handle it by cache and pipelining.
>
> Thank you for the pointer. We will discuss this related work in the revision. As you noted, if the f‑gram embedding table were to be fully materialized during training, techniques such as caching and pipelining could help mitigate the latency issue. However, beyond latency, a key challenge with materializing the f‑gram embedding table is the problem of sparse updates. Due to the heavy-tail distribution of natural language, many f‑gram embeddings would receive only a handful of gradient updates per epoch. As we show in Appendix D of the submission, this sparse update issue can eventually degrade model performance. In contrast, when f‑gram representations are generated through a neural network, the parameter updates are shared across all f‑grams, which alleviates this problem.
>
>
> **6.2** Page 3"During training, it is parameterized by ...". This sentence is confusing.
>
> We have revised the corresponding sentence to:
>
> “During training, instead of storing an explicit embedding table for f-grams, we generate their embeddings on the fly. A separate transformer model, $\mathcal{A}_{\text{f-gram}}$, takes an f-gram as input, and the embedding of its final token is used as the representation of that f-gram.”  Hope this is clearer.
>
> **6.3** Page 5 "... instantiate a full embedding table during training" would be much better if you add this is the f-gram table.
>
> We have revised accordingly.

---

> > ### Comment · Reviewer_yxCy · 2025-08-05
> >
> > Thanks for the response.

---

### Official Review · Reviewer_rTKx · 2025-07-01

**Clarity:** 3
**Significance:** 4
**Originality:** 4
**Rating:** 5
**Confidence:** 4

**Summary:**

This paper introduces SCONE, a novel and practical method for scaling language models by enhancing the input embedding layer. Instead of simply enlarging the token vocabulary, which has known drawbacks, the authors propose augmenting it with embeddings for frequently occurring n-grams (f-grams). These f-gram embeddings are learned by a separate, smaller transformer model during training. Critically, for inference, these embeddings are precomputed and offloaded to cheaper, more abundant resources like system memory or SSDs. This design allows the model to leverage richer, contextualized input representations to improve performance without incurring additional computational (FLOPS) or memory costs on the accelerator during inference. The experiments convincingly demonstrate that this approach enables smaller models to outperform significantly larger baselines,

**Questions:**

Please refer to my weakness part.

**Ethical Concerns:**

["NO or VERY MINOR ethics concerns only"]

**Limitations:**

yes

**Paper Formatting Concerns:**

N/A.

**Quality:**

4

**Strengths And Weaknesses:**

**Strengths:**

- The paper's core idea is highly innovative. Much of the research on LLM architecture focuses on attention and FFN layers, while the input embedding layer is often overlooked.  SCONE provides a fresh perspective by demonstrating that this seemingly simple component can be a powerful lever for scaling. The approach of decoupling the learning of contextual n-grams (during training) from their application (a simple lookup during inference) is both clever and effective.

- The experimental validation is thorough and convincing.  The authors conduct systematic experiments across multiple model sizes and datasets, including well-designed ablation studies on key hyperparameters like the number of f-grams and the f-gram model size. The initial reliance on perplexity was a concern, but the authors' rebuttal comprehensively addressed this by providing extensive results on standard downstream benchmarks, which strongly reinforce their claims. The paper is well-written, and the core concept is simple and easy to grasp.

**Weaknesses:**
- The paper focuses exclusively on offloading computations related to input n-grams. Transformers implicitly learn to process n-gram patterns in their attention and feedforward layers.  A fascinating future direction would be to investigate if other frequent computational patterns within the main model could also be identified, precomputed, and offloaded to further improve efficiency.
- While the empirical results are compelling, a deeper theoretical analysis of the trade-offs would be beneficial. For instance, quantifying the expressiveness added by the f-gram embeddings versus the computational cost of the main model could provide a more formal understanding of SCONE's scaling properties.

---

> ### Author Rebuttal · Authors · 2025-07-30
>
> We sincerely thank the reviewer for their insightful review. Please find our responses below. We would be happy to address any further questions or clarifications if needed.
>
> **1. A fascinating future direction would be to investigate if other frequent computational patterns within the main model could also be identified, precomputed, and offloaded to further improve efficiency.**
>
> We completely agree that identifying and caching other frequent computational patterns within the main model is a fascinating direction for future work. In Appendix A of our submission, we discuss two interesting challenges in extending caching beyond n-gram embeddings: (1) Using raw text as cache keys can lead to low hit rates as the sequence length increases, and (2) using semantic embeddings as keys would require carefully designed discretization techniques to map continuous embeddings into a discrete key space that also supports efficient indexing.
>
> **2. A deeper theoretical analysis of the trade-offs would be beneficial. For instance, quantifying the expressiveness added by the f-gram embeddings versus the computational cost of the main model could provide a more formal understanding of SCONE's scaling properties.**
>
> While the primary focus of this work is on proposing a new method and conducting an empirical study, we agree that a deeper theoretical analysis would be a valuable direction for future research. We will acknowledge this in the “Limitations and Future Work” section of the revision.  However, please note that providing a formal theoretical characterization of modern LLMs has been challenging, even without SCONE.
>
> **3. New results on post-training of Qwen3 models**
>
> In addition to addressing the reviewer’s comments, we conducted new experiments during the rebuttal period to demonstrate that SCONE can also be applied at the post-training stage of the latest Qwen3 models. Although these experiments are not directly related to your specific questions, we share them here to further illustrate the broad applicability of SCONE.
>
> Specifically, we applied SCONE to supervised fine-tuning (SFT) of [Qwen3-4B-base](https://huggingface.co/Qwen/Qwen3-4B-Base) using the [open-r1](https://github.com/huggingface/open-r1) framework and the [open-r1/Mixture-of-Thoughts](https://huggingface.co/datasets/open-r1/Mixture-of-Thoughts) dataset. In this setup, Qwen3-4B serves as the main model, while Qwen3-8B-base or Qwen3-14B-base are used as the f-gram models. We set the number of f-grams to 10M and use the same training hyperparameters as open-r1. The table below compares the resulting SCONE-enhanced models with the Qwen3-4B baseline in terms of both accuracy and decoding latency.
>
> |   |  AIME 2024 pass@1  |  LiveCodeBench v4_v5 pass@1 | Decoding Latency at inference (per-token) |
> |----------|----------|----------|----------|
> | Qwen3-4B-base Baseline| 45.3 | 30.8 | 10.05 ms |
> |Qwen3-4B SCONE (Qwen3-8B f-gram model) | 48.3 (+3.0) | 34.5 (+3.7) | 10.13 ms |
> |Qwen3-4B SCONE (Qwen3-14B f-gram model) | 51.6 (+6.3)  | 36.3 (+5.5)  | 10.13 ms |
>
> These results show that SCONE consistently improves performance over the baseline, with larger f-gram models yielding greater gains, all while maintaining similar inference latency.

---

> > ### Comment · Reviewer_rTKx · 2025-08-05
> >
> > Thank you for your extensive experiments, I will keep my initial score.

---

> > > ### Author Response · Authors · 2025-08-05
> > >
> > > Thank you again for your thoughtful review and for taking the time to read our response!

---

### Official Review · Reviewer_NngF · 2025-07-03

**Clarity:** 3
**Significance:** 3
**Originality:** 2
**Rating:** 4
**Confidence:** 4

**Summary:**

This paper introduces SCONE, a novel method for improving language models by scaling the input embedding layer without increasing inference-time accelerator costs. The core idea is to augment the standard token embeddings with contextualized embeddings for a large set of frequent n-grams (termed "f-grams"). To avoid the challenges of training a massive, static embedding table, SCONE employs a separate "f-gram model" during training to learn and generate these embeddings dynamically. After training, these f-gram embeddings are precomputed and stored in an off-accelerator lookup table (e.g., in main memory or on an NVMe drive), which is queried efficiently during inference. This design decouples the input embedding size from the main model's vocabulary and output layer, sidestepping the computational bottlenecks of traditional vocabulary scaling.

**Questions:**

* Interaction with Mixture-of-Experts (MoE) Architectures: The paper positions SCONE as an alternative scaling approach to MoE, highlighting SCONE's advantage in maintaining a fixed accelerator memory footprint. However, given that MoE is a prevalent architecture for state-of-the-art large language models, it would be valuable to explore the interaction between these two methods. Could SCONE and MoE be used together, and if so, are their benefits complementary or do they overlap? For instance, would applying SCONE to an MoE model yield additive performance gains, or would one method's benefits diminish the other's? An experimental analysis or at least a discussion on this potential synergy would be highly relevant.
* Domain Adaptation and Continual Learning: The current approach relies on a static f-gram set defined at the start of pre-training. A key practical question is how the model adapts when moved to a new domain (e.g., legal or medical texts). A discussion or experiment on how to efficiently integrate new, high-frequency n-grams from a new domain would significantly strengthen the paper's contribution to real-world applicability.

**Ethical Concerns:**

["NO or VERY MINOR ethics concerns only"]

**Limitations:**

Yes

**Quality:**

3

**Strengths And Weaknesses:**

Strength:

* The idea of training an f-gram model is an good approach to address the challenges of training large-scale, sparse vocabularies.
* The paper is well-written, clear, and easy to understand.
* The work demonstrates two effective scaling dimensions: increasing the size of the f-gram vocabulary and scaling up the f-gram model.

Weakness:

* Clarity on Training and Pre-computation Costs: While the authors make a commendable effort to create a fair comparison by reducing the number of training tokens for SCONE models to approximate equivalent training FLOPS (as mentioned in Section 4.2), the comparison between the SCONE-enhanced 1B model and the 1.9B baseline could be further clarified. The training dynamics of a smaller model trained for more steps versus a larger, more complex architecture trained for fewer steps can be different. A more direct comparison under a fixed total training time or cost budget would help establish whether the architectural complexity of SCONE offers a definitive advantage over simply training a larger, monolithic model, which offers a simpler architecture. Furthermore, the paper could benefit from a more detailed analysis of the pre-computation costs. For instance, quantifying the time and computational resources required for the 1.8B f-gram model to generate the 1B embeddings would provide a more complete picture of the total resource footprint of the SCONE methodology.
* Scalability Concerns in High-Concurrency Deployment: The paper provides a valuable analysis of the query latency for the offloaded f-gram embedding table (Section 4.3), demonstrating its feasibility in single-instance, batched inference scenarios. However, the experiments are conducted on a single workstation with batch sizes up to 16. It would strengthen the paper to discuss the potential engineering challenges and performance implications in a large-scale, production deployment. In a real-world serving environment with high concurrency (i.e., many simultaneous, independent user requests), the contention for system memory bandwidth or NVMe I/O from these frequent lookups could potentially become a bottleneck. A discussion on how the system would perform under such high-throughput, multi-tenant conditions would add significant practical depth to the evaluation.

---

> ### Author Rebuttal · Authors · 2025-07-30
>
> Thank you for your review! We have incorporated your suggestions to improve our manuscript. Please find our responses below.
>
> **1. Clarity on Training and Pre-computation Costs**
>
> **1.1** The comparison between the SCONE-enhanced 1B model and the 1.9B baseline could be further clarified.
>
> We compare the training dynamics of the 1.9B baseline with the SCONE-enhanced 1B model (which uses a 1.8B f-gram model and 1B cached f-gram embeddings).  Since we cannot share images via external links, we summarize the results in the following tables. The first table shows the training FLOPs per sequence (for SCONE averaged across sequences), and the second shows evaluation perplexity at comparable training FLOPs.
>
> **Training FLOPS / sequence**
>
> | SCONE 1B (1.8B f-gram model) | 1.9B baseline |
> |----------|----------|
> | $5.57\times 10^{13}$| $2.73\times 10^{13}$ |
>
> **Evaluation perplexity vs. training FLOPs**
>
> | | $\sim 2.66\times 10^{21}$  | $\sim 5.32\times 10^{21}$ | $\sim 7.98\times 10^{21}$ | $\sim 1.06\times 10^{22} $
> |----------|----------|----------|----------|----------|
> | 1.9B baseline | 12.680 | 11.246  |  10.570 | 10.085 |
> | SCONE 1B (1.8B f-gram model) | 12.598 | 11.256 | 10.524 | 10.033 |
>
> The SCONE-enhanced 1B model achieves performance comparable to the 1.9B baseline when matched on training FLOPs, while requiring roughly half the FLOPs at inference. We will add the corresponding plots to the revision.
>
>
> **1.2** The paper could benefit from a more detailed analysis of the pre-computation costs.
>
> The FLOPs and processing time for pre-computation on a node with 8 A100 GPUs are shown below:
>
>
> | Total FLOPS | 	Processing time |
> |----------|----------|
> | $\sim 1.836\times 10^{17}$| 2 hour 35 mins |
>
> The cost of pre-computing 1 billion f‑gram embeddings (requiring forward passes over 1B short n‑grams) is negligible compared to pre-training, which involves processing 1T tokens over full sequence lengths. We will add a discussion of these pre-computation costs to the revision.
>
>
> **2. A discussion on how the system would perform under such high-throughput, multi-tenant conditions would add significant practical depth to the evaluation.**
>
> Thank you for the insightful question. While a batch size of 16 is already larger than what is typically used for model inference in production (since accelerator memory usage grows linearly with batch size and decoded sequence length), we agree that real-world deployments often operate in high-throughput, multi-tenant environments. In such scenarios, a common approach is to replicate the serving instances so that each replica handles at most the target batch size of concurrent queries. For SCONE, this implies that the f-gram embedding table would need to be replicated across physical devices. However, we note that this requirement is not unique to SCONE: traditional LLM serving pipelines also need to replicate model parameters across servers to handle high concurrency.
>
>
> **3. Could SCONE and MoE be used together, and if so, are their benefits complementary or do they overlap?**
>
> While SCONE and MoE can easily be combined since SCONE only modifies the input embeddings of the main model, we acknowledge that our current experiments do not include results with MoE models. This is primarily because pre-training large MoE models requires substantial computational resources, which is beyond the scope of our current work. We do, however, discuss the conceptual connections between SCONE and MoE in Appendix B of our submission, and we will revise to explicitly address this point in the Limitations section as well.
>
> To address the question raised by Reviewer 5vFb, we have conducted additional experiments using the recent Qwen3 4B/8B/14B models at the **post-training stage**, which is considerably less resource-intensive than pre-training. Although these Qwen models are not MoE, we believe the results provide **partial evidence for the broader applicability of SCONE across different model architectures**.
> Specifically, we apply SCONE during supervised fine-tuning (SFT) on [Qwen3-4B-base](https://huggingface.co/Qwen/Qwen3-4B-Base), following the [open-r1](https://github.com/huggingface/open-r1) framework, while using Qwen3-8B-base or Qwen3-14B-base as the f-gram model. These experiments use the [open-r1/Mixture-of-Thoughts](https://huggingface.co/datasets/open-r1/Mixture-of-Thoughts) SFT dataset, with 10M f-grams and the same training hyperparameters as open-r1. The table below summarizes the results, comparing SCONE-enhanced models against the Qwen3-4B baseline in terms of accuracy and inference latency:
>
>
>
> |   |  AIME 2024 pass@1  |  LiveCodeBench v4_v5 pass@1 | Decoding Latency at inference (per-token) |
> |----------|----------|----------|----------|
> | Qwen3-4B-base Baseline| 45.3 | 30.8 | 10.05 ms |
> |Qwen3-4B SCONE  (Qwen3-8B f-gram model) | 48.3 (+3.0) | 34.5 (+3.7) | 10.13 ms |
> |Qwen3-4B SCONE  (Qwen3-14B f-gram model) | 51.6 (+6.3)  | 36.3 (+5.5)  | 10.13 ms |
>
> These results demonstrate that SCONE remains effective when applied to state-of-the-art architectures, with negligible impact on decoding latency.
>
> **4. A discussion or experiment on how to efficiently integrate new, high-frequency n-grams from a new domain would significantly strengthen the paper's contribution to real-world applicability.**
>
> We construct the f-gram set from a large and diverse pre-training corpus. The zero-shot evaluation results in Section 4.2 (covering math, coding, factuality QA, etc.) indicate that this f-gram set generalizes well across a wide range of tasks. That said, we agree that it would be valuable for the f‑gram set and its embeddings to adapt to new domains that are not presented in pre-training data, and we will include this point in the Limitations section. To partially address this concern, the new post-training results suggest that SCONE can also be applied after pre‑training, allowing the f‑gram embedding table to be updated without repeating the entire pre‑training process.

---

### Note · Authors · 2025-08-12

Dear AC and reviewers,

We thank the AC and reviewers for their careful handling of our paper and the constructive discussion process. We appreciate the chance to provide these final remarks, which summarize: (1) main strengths noted by reviewers, (2) new analyses from the discussion period, and (3) acknowledgements of remaining limitations.

**1. Main strengths noted by reviewers**

Below we summarize several shared strengths (space constraints prevent us from listing all):

- Clear writing: “Well-written, clear, and easy to understand” (NngF); “extremely well-written” (yxCy); “clear writing and logic” (H6nF).
- Novelty and motivation: “A good approach to address large-scale, sparse vocabularies” (NngF); “highly innovative” (rTKx); “natural and well-justified design choices” (yxCy).
- Comprehensive experiments: “Experimental validation is thorough and convincing” (rTKx); “demonstrates two effective scaling dimensions” (NngF); “detailed scaling results” (5vFb); “effective performance improvement” (H6nF).

**2. New analyses during discussion**

We presented additional results that will be included in the revised manuscript:

- Training cost clarity (NngF): Added more FLOPS comparisons between SCONE and baselines, plus perplexity curves vs. FLOPS scaling.
- Tradeoff summary (yxCy): Added a table of GPU/CPU memory, disk usage, training FLOPS, and decoding latency.
- Post-training applicability (5vFb, H6nF, NngF): Applied SCONE in supervised fine-tuning of larger Qwen3-base models, demonstrating adaptability to math and code reasoning tasks.

**3. Acknowledged limitations**

Although no reviewers expressed further concerns after the discussion period, we will discuss the following in the Limitations section:

- Larger scale pre-training: We only pre-trained models up to 3B parameters due to resource limits, though larger Qwen3 models showed positive post-training results.
- Broader architecture coverage: We have not evaluated SCONE with Mixture-of-Experts, Mamba, or GLA architectures. While models we used are common choices, exploring these alternatives may reveal new insights.
- Theoretical analysis: Our work demonstrates SCONE’s broad empirical applicability but lacks a deep theoretical understanding. Developing such an analysis remains important, though formal characterization of modern LLMs is challenging even without SCONE.


We hope this summary supports your decision-making, and we thank you again for the valuable feedback and engagement.

---

### Decision · Program_Chairs · 2025-09-17

**Decision:**

Accept (poster)

**Comment:**

The paper received strong evaluations. It is well-written, has clear motivation, and the design choices are well-justified. I recommend that the authors include additional experiments conducted during the rebuttal and the limitations of their work (which they presented in the final remarks) in the camera-ready version of the paper.